# Epigenetic regulation of hematopoiesis by DNA methylation

Aniket V Gore[1], Brett Athans[1], James R Iben[2], Kristin Johnson[1], Valya Russanova[1], Daniel Castranova[1], Van N Pham[1], Matthew G Butler[1], Lisa Williams-Simons[1], James T Nichols[3], Erica Bresciani[4], Bejamin Feldman[1], Charles B Kimmel[3], Paul P Liu[4], Brant M Weinstein[1]*

[1]Division of Developmental Biology, National Institute of Child Health and Human Development, National Institutes of Health, Bethesda, United States; [2]Program in Developmental Endocrinology and Genetics, National Institute of Child Health and Human Development, National Institutes of Health, Bethesda, United States; [3]Institute of Neuroscience, University of Oregon, Eugene, United States; [4]Oncogenesis and Development Section, National Human Genome Research Institute, National Institutes of Health, Bethesda, United States

**Abstract** During embryonic development, cell type-specific transcription factors promote cell identities, while epigenetic modifications are thought to contribute to maintain these cell fates. Our understanding of how genetic and epigenetic modes of regulation work together to establish and maintain cellular identity is still limited, however. Here, we show that *DNA methyltransferase 3bb.1* (*dnmt3bb.1*) is essential for maintenance of hematopoietic stem and progenitor cell (HSPC) fate as part of an early *Notch-runx1-cmyb* HSPC specification pathway in the zebrafish. Dnmt3bb.1 is expressed in HSPC downstream from *Notch1* and *runx1*, and loss of Dnmt3bb.1 activity leads to reduced *cmyb* locus methylation, reduced *cmyb* expression, and gradual reduction in HSPCs. Ectopic overexpression of *dnmt3bb.1* in non-hematopoietic cells is sufficient to methylate the *cmyb* locus, promote *cmyb* expression, and promote hematopoietic development. Our results reveal an epigenetic mechanism supporting the maintenance of hematopoietic cell fate via DNA methylation-mediated perdurance of a key transcription factor in HSPCs.

*For correspondence: WeinsteB@mail.nih.gov

**Competing interests:** The authors declare that no competing interests exist.

## Introduction

Generating and maintaining stable cell fates is crucial for normal development. Many of the key cell types generated during embryonic development must be maintained throughout adult life to support normal organ and tissue function and homeostasis. Hematopoietic stem and progenitor cells (HSPCs) are formed during early development and are responsible for maintaining the lifelong supply of blood cells, but the factors necessary for their generation and maintenance are still not well understood. HSPCs arise during embryogenesis from hemogenic endothelial cells located in the ventral floor of the trunk dorsal aorta (*Bertrand et al., 2010a*; *Kissa and Herbomel, 2010*; *Boisset et al., 2010*; *Murayama et al., 2006*) that enter the circulation to seed hematopoietic organs and tissues and eventually give rise to all differentiated blood lineages (*Bertrand et al., 2010a*; *Kissa and Herbomel, 2010*; *Murayama et al., 2006*). The emergence of HSPCs from aortic endothelium is regulated by a Notch-Runx1-Cmyb signaling pathway (*Burns et al., 2005*; *Clements et al., 2011*; *Richard et al., 2013*; *Bertrand et al., 2010b*; *Rowlinson and Gering, 2010*). Studies in mice and zebrafish have shown that the Runx1 transcription factor is required downstream from Notch signaling and upstream from Cmyb for HSPC development from the endothelium, although once HSPCs are specified, Runx1 transcription is down-regulated and it is not necessary for

**eLife digest** The cells in our blood are constantly being replaced with new cells that are produced by stem cells called hematopoietic stem and progenitor cells (or HSPCs for short). The HSPCs form early on in the development of the embryo and continue in the same role throughout the life of the animal.

A gene called *runx1* is required for HSPCs to form, but is not required for these cells to maintain their role (cell identity) in the long term. In mice, this gene is only expressed for a brief period of time as the HSPCs form, and is switched off in the mature stem cells. Another gene called *cmyb* – which is switched on by *runx1* – is also required for HSPCs to form. However, unlike *runx1*, *cmyb* continues to be expressed in mature HSPCs and is required to maintain HSPC identity. It is not known how the temporary activation of *runx1* causes the long-term expression of *cmyb*.

One possible explanation is that the *cmyb* gene may be subject to a process called DNA methylation. This process is carried out by enzymes called DNA methyltransferases and can have long-term effects on the expression of genes by modifying the structure of the DNA that encodes them. Here, Gore et al. investigate the role of a particular DNA methyltransferase in the formation of HSPCs in zebrafish embryos. The experiments show that this enzyme is activated in developing HSPCs in response to an increase in *runx1* expression. The loss of this enzyme's activity reduces both the amount that *cmyb* is methylated and its level of expression, which results in a gradual decline in the number of HSPCs in zebrafish.

Further experiments show that if the DNA methyltransferase is artificially activated in cells that don't normally form blood cells, these cells change their identity to do so. This switch is accompanied by methylation of *cmyb* and an increase in its expression. Gore et al.'s findings reveal that the temporary activation of *runx1* triggers the production of an enzyme that methylates *cmyb* to maintain the identity of HSPCs. Future studies should help to reveal exactly how *runx1* promotes DNA methylation, and whether this process can be harnessed to promote HSPC formation for research or medical treatments.

HSPC fate maintenance (*Chen et al., 2009*; *Liakhovitskaia et al., 2009*; *Sood et al., 2010*; *Kissa and Herbomel, 2010*; *Tober et al., 2013*; *Lam et al., 2009*). The transcription factor Cmyb is also required for HSPC formation downstream from Runx1. However, unlike Runx1, Cmyb expression persists in HSPC and its function is required not only for differentiation of HSPCs but also for their subsequent maintenance (*Mukouyama et al., 1999*; *Mucenski et al., 1991*; *Soza-Ried et al., 2010*; *Zhang et al., 2011*). It is not clear how transient expression of Runx1 leads to the long-lasting expression and persistent functional role for Cmyb and maintenance of HSPC identity. Epigenetic regulatory mechanisms have been implicated in long-term maintenance of gene expression (*Deaton and Bird, 2011*) and differentiated cell fate (*Hu et al., 2012*; *Mohn et al., 2008*; *Lay et al., 2015*; *Wu et al., 2010*), and we hypothesized that they might play a role in HSPC fate maintenance during the Runx1-independent phase.

A number of different mechanisms have been described for epigenetic regulation of gene expression, including post-translational modification of histones by the covalent addition of acetyl, methyl, and other groups to specific histone amino acid residues (*Greer and Shi, 2012*). DNA methylation is another well-studied and important mechanism for epigenetic gene regulation (*Suzuki and Bird, 2008*). In eukaryotes, DNA methyltransferases (DNMTs) add a methyl group to the 5 position of cytosine residues in DNA, typically on cytosines in CpG dinucleotides. CpGs are often grouped in clusters called 'CpG islands' (CGIs), which are themselves frequently found within or in the 5' regions of genes (*Deaton and Bird, 2011*; *Suzuki and Bird, 2008*). DMNTs are classified as either 'maintenance' DNMTs, responsible for preserving existing DNA methylation after every cellular DNA replication cycle, or 'de novo' DNMTs, which place new methyl 'marks' on DNA. Mammalian DNMT1 acts as a maintenance enzyme and has high affinity for binding to hemimethylated DNA. Other mammalian DNMTs, including the DNMT3 family enzymes DNMT3A and DNMT3B, act as de novo methyltransferases and establish initial DNA methylation patterns (*Goll and Bestor, 2005*; *Xu et al., 2010*; *Cheng and Blumenthal, 2008*). Genomic imprinting by DNA methylation has a well-

documented role in reducing gene expression in both plants and animals (*Reik and Dean, 2001*), for example, during X-chromosome inactivation in mammals (*Tada et al., 2000*; *Gendrel and Heard, 2014*) or repression of imprinted allele expression during Arabidopsis embryonic development (*Jullien and Berger, 2009*). DNA methylation can also act as a positive regulator for gene expression, particularly when methylation occurs in gene bodies as opposed to promoter or enhancer regions (*Baubec et al., 2015*; *Yang et al., 2014*). In mouse stem cells, DNMT3B selectively binds to and methylates gene body regions, excluding promoter and enhancer regions. DNMT3B binding is found preferentially in gene body regions of actively expressed genes (*Baubec et al., 2015*). In a recent elegant study performed in the chick, DNMT3A was shown to act as an epigenetic switch repressing neural and promoting neural crest fate by binding to and methylating the SOX2 and SOX3 promoters and inhibiting their expression (*Hu et al., 2012*). These findings highlight the important role that epigenetic regulation plays in establishing cell fate and the diverse functions DNMTs play in regulating gene expression during early embryonic development.

Here, we examine the molecular mechanisms responsible for maintaining HSPC fate independent of Runx1 function in the zebrafish. We find that the de novo DNA methyltransferase Dnmt3bb.1 functions downstream from Runx1 to maintain cmyb gene expression and HSPC cell fate. Our findings reveal a previously unknown epigenetic mechanism regulating HSPC fate maintenance.

## Results

### Dnmt3bb.1 is expressed by developing HSPCs

Using whole-mount in situ (WISH) hybridization, we find that around 36 hpf selected cells in the developing zebrafish dorsal aorta specifically express DNA methyltransferase 3bb.1 (*dnmt3bb.1*) (*Figure 1a,b*), a gene most closely related to human DNMT3B (*Figure 1—figure supplement 1a*). As noted above, DNMT3B functions as a de novo DNMT in humans, adding new methyl 'marks' to cytosine residues in DNA. Dnmt3bb.1-positive cells are present in the ventral floor of the dorsal aorta (*Figure 1c,d*) (*Takayama et al., 2014*; *Thisse, 2001*)), and double WISH confirms that *dnmt3bb.1* is co-expressed with *cmyb* (*Figure 1e,f*), a known marker for developing HSPCs (*Gering and Patient, 2005*). *dnmt3bb.1* is expressed specifically in HSPCs in the developing trunk, although expression is also present in portions of the eye and a few other selected head tissues during early development (*Aanes et al., 2011*) (*Figure 1—figure supplement 1b–e*). As noted above, a Notch-Runx1-Cmyb pathway regulates the endothelial to HSPC transition (*Figure 1g*) (*Burns et al., 2005*; *Chen et al., 2009*; *Kissa and Herbomel, 2010*). To determine whether *dnmt3bb.1* acts in conjunction with this pathway, we examined the expression of *dnmt3bb.1* using WISH and quantitative reverse transcriptase-polymerase chain reaction (qRT-PCR) after manipulating *runx1* or *notch* (*Figure 1h–m*). Expression of *dnmt3bb.1* is strongly reduced in *runx1* morpholino-injected animals or *runx1^{W84X}* mutants (*Sood et al., 2010*) (*Figure 1h,i,m*), while over-expression of *runx1* by injection of *runx1* mRNA results in increased expression of *dnmt3bb.1* in the axial vasculature (*Figure 1i,j,m*). Similarly, expression of *dnmt3bb.1* is also strongly reduced in Notch-deficient *mindbomb (mib)* mutants (*Figure 1k,l,m*). These results show that *dnmt3bb.1* is expressed by developing HSPCs downstream from the established Notch-Runx1 pathway for HSPC specification (*Burns et al., 2005*; *Chen et al., 2009*; *Kissa and Herbomel, 2010*), and that its expression is lost when HSPCs are not properly specified.

### Dnmt3bb.1 is required for hematopoiesis

To test whether Dnmt3bb.1 function is necessary for hematopoietic development, we generated genetic mutants in the *dnmt3bb.1* locus using TALEN technology (*Dahlem et al., 2012*). We obtained a *dnmt3bb.1^{y258}* mutant allele encoding a polypeptide prematurely truncated at amino acid 172 (P166fs) due to an 11 nucleotide deletion (*Figure 2—figure supplement 1a–e*). Morphologically, two to five dpf *dnmt3bb.1^{y258}* mutants are indistinguishable from their wild-type siblings (*Figure 2a,b*). At 36 hpf, *cmyb* expression in HSPCs in the ventral aorta is similar in *dnmt3bb.1^{y258}* mutant embryos and their wild type siblings (*Figure 2c,d*). However, by 72 and 96 hpf *cmyb* expression is strongly reduced in the caudal hematopoietic tissue (CHT) of *dnmt3bb.1^{y258}* mutants (*Figure 2e–h*), as confirmed by genotyping of blindly scored embryos (*Figure 2—figure supplement 1e*). Loss of *cmyb* is also accompanied by strong reduction in myeloid marker *l-*

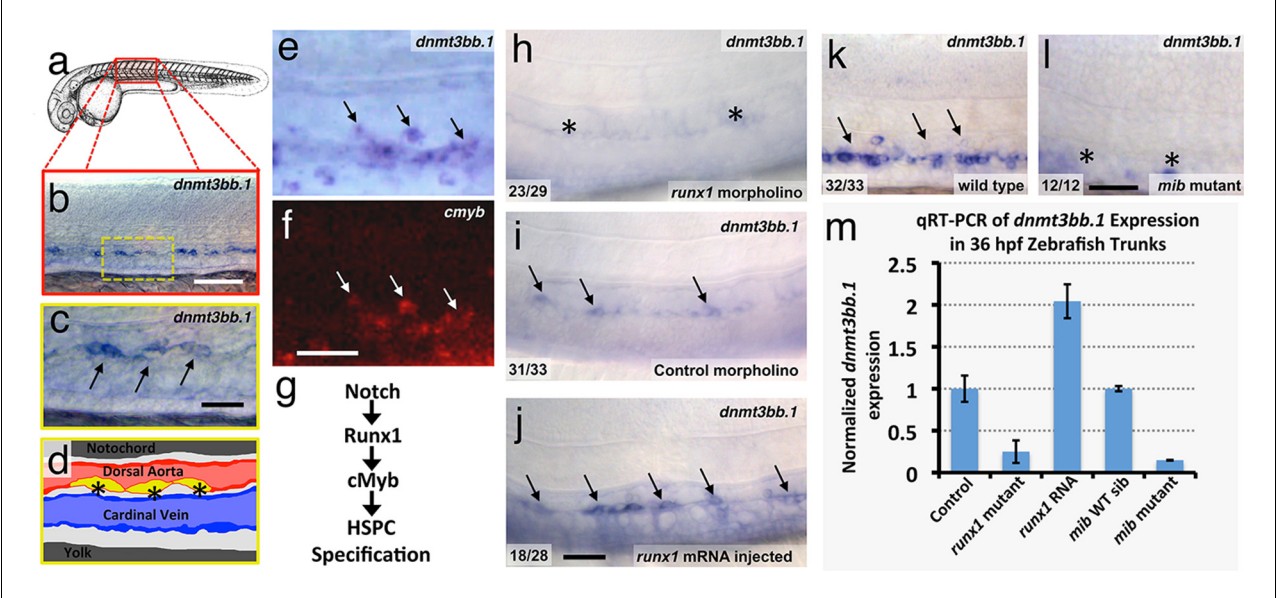

**Figure 1.** The DNA methyltransferase3bb.1 gene is expressed in developing hematopoietic stem and progenitor cells and is regulated by hematopoietic stem and progenitor cell ( HSPC)-specific pathways. (a) Camera lucida drawing of a 24 hpf zebrafish embryo with a red box noting the approximate region of the trunk shown in in situ hybridization images. (b,c) Whole-mount in situ hybridization of a 36 hpf zebrafish trunk probed for *dnmt3bb.1*, showing expression in the ventral floor of the dorsal aorta (arrows in panel c). Yellow inset box in panel b indicates the magnified area shown in panel c. (d) Diagram corresponding to panel c showing dorsal aorta (red), cardinal vein (blue), and *dnmt*-positive cells in the floor of the dorsal aorta (yellow). (e,f) Double in situ hybridization of a 36 hpf zebrafish trunk probed for *dnmt3bb.1* (e, blue) and *c-myb* (f, red). The *c-myb* positive HSPC progenitors also stain for *dnmt3bb.1* (arrows). (g) A Notch- Runx- c-myb pathway regulates HSPC emergence in the zebrafish. (h–j) In situ hybridization of 36 hpf *runx1* morpholino-injected (h), control (i), or *runx1* mRNA-injected (j) zebrafish trunks probed for *dnmt3bb.1,* showing that *dnmt3bb.1* expression is reduced by *runx1* knockdown (asterisks) and increased by *runx1* overexpression (arrows). (k,l) In situ hybridization of 36 hpf wild type (k) or *mind bomb* mutant (l) zebrafish trunks probed for *dnmt3bb.1,* showing that *dnmt3bb.1* expression seen in wild-type siblings (arrows in panel k) is strongly reduced in Notch-deficient *mind bomb* mutants (asterisks in panel l). (m) Quantitative reverse transcriptase-polymerase chain reaction (qRT-PCR) analysis of *dnmt3bb.1* transcript levels in 36 hpf (i) control, (ii) *runx1*[W84X] mutant, (iii) *runx1* mRNA injected, (iv) *mind bomb (mib)* wild type sibling (wild type sibling of *mib* mutants), and (v) *mib* mutant zebrafish embryos. *dnmt3bb.1* levels are normalized to the reference gene *elf1α* and to levels in controls. All graphs show mean ± standard error of the mean (SEM) and are representative of three biological replicates. Scale bars = 100 µm in b, h-l and 50 µm in c, f.

The following figure supplement is available for figure 1:

**Figure supplement 1.** Dnmt3bb.1expression analysis and homology.

plastin (**Herbomel et al., 1999**) in the CHT at 72 hpf (**Figure 2i,j**) and lymphoid marker *rag1* (**Willett et al., 1997**) in the 5 dpf thymus (**Figure 2k,l**). We observed similar hematopoietic defects in wild type embryos injected with any one of three different dnmt3bb.1-targeting morpholinos, with progressive loss of *cmyb* expression (**Figure 2—figure supplement 2a–f**) as well as reduced expression of *l-plastin* (**Figure 2—figure supplement 2g,i**) and erythroid marker *gata1* (**Figure 2—figure supplement 2k–n**) in the 72 hpf CHT, and reduced expression of *rag1* (**Figure 2—figure supplement 2h,j**), *ikaros*, and *lck:gfp* (**Figure 2—figure supplement 2o–r**) in the 5 dpf thymus. Quantitative RT-PCR confirmed progressive loss of *cmyb* and strong reduction in subsequent *l-plastin* and *rag1* expression in Dnmt3bb.1-deficient animals (**Figure 2m,n**). Strong increase in active caspase 3 in HSPC in the trunks of 48 hpf Dnmt3bb.1-deficient animals indicated that HSPCs are undergoing apoptosis (**Figure 2—figure supplement 3a–c**). Despite dramatic hematopoietic defects, Dnmt3bb.1-deficient animals appeared otherwise unaffected, with normal vascular patterning and vascular gene expression (**Figure 2—figure supplement 3d–i**). These results suggest that loss of *dnmt3bb.1* activity leads to a specific defect in HSPCs, with progressive loss of *cmyb* expression and decreased numbers of HSPCs.

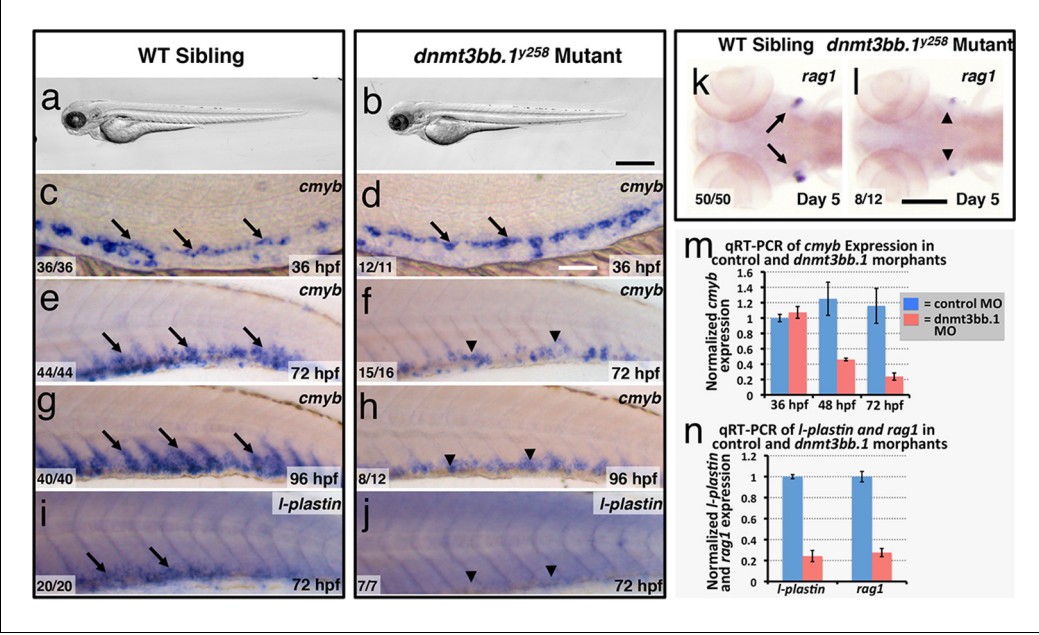

**Figure 2.** Dnmt3bb.1is necessary for hematopoietic gene expression. (**a,b**) Transmitted light images of 72 hpf control (a) and *dnmt3bb.1*[y258] (b) mutant zebrafish, showing absence of developmental delay or gross morphological abnormalities in *dnmt3bb.1*[y258] mutants. (**c–h**) Whole mount in situ hybridization of 36 hpf (c,d), 72 hpf (e,f) and 96 hpf (g,h) wild type (WT) sibling (c,e,g) or *dnmt3bb.1*[y258] (d,f,h) mutant animals, probed for *cmyb*. *Cmyb* is expressed in *dnmt3bb.1* mutant at 36 hpf (arrows) but this expression is strongly reduced by 72 and 96 hpf (arrowheads). (**i–l**) Whole mount in situ hybridization of 72 hpf tails probed for *l-plastin* (i,j) and 5 dpf heads probed for *rag1* (k,l) from WT sibling (i,k) or dnmt3bb.1 (j,l) mutant animals. Expression of both *l-plastin* and *rag1* is present in controls (arrows) but strongly reduced in *dnmt3bb.1* mutant animals (arrowheads). (**m**) Quantitative RT-PCR analysis of *cmyb* transcript levels in control (blue columns) or dnmt3bb.1 (red columns) morpholino injected animals at 36, 48, and 72 hpf. Transcript levels are normalized to the reference gene *elf1α* and to levels in 36 hpf control morphants. (**n**) Quantitative RT-PCR analysis of 3 dpf *l-plastin* and 5 dpf *rag1* transcript levels in control (blue columns) or dnmt3bb.1 (red columns) morpholino-injected animals. Transcript levels are normalized to the reference gene *elf1α* and to levels in controls. All graphs in panels i and j show mean ± SEM, are representative of three biological replicates. Scale bars = 150 μm in a,b, 50 μm in c,d, 100 μm in e-l.

The following figure supplements are available for figure 2:

**Figure supplement 1.** TALENmediated Dnmt3bb.1 mutation.

**Figure supplement 2.** Dnmt3bb.1 morpholino phenocopies TALEN-induced mutant defects.

**Figure supplement 3.** HSPCapoptosis but no gross morphological and vascular defects in in *dnmt3bb.1* morphants.

## Gene expression and DNA methylation changes in Dnmt3bb.1-deficient HSPC

To examine whether changes in DNA methylation correlate with hematopoietic defects resulting from the loss of dnmt3bb.1, we isolated HSPCs from 36 hpf control or dnmt3bb.1-deficient animals using a previously reported double-transgenic Fluorescence Activated Cell Sorting (FACS) method (*Bertrand et al., 2010a*) (*Figure 3a*). DNA and RNA were prepared simultaneously from HSPC-enriched samples for global analysis of DNA methylation and gene expression using Reduced Representation Bisulfite Sequencing (RRBS) (*Meissner et al., 2005*) and RNAseq, respectively (*Figure 3a*, *Figure 3—figure supplement 1a*). RNAseq analysis (and subsequent qPCR validation of selected genes) showed that the expression of a variety of hematopoietic genes was reduced in dnmt3bb.1-deficient HSPCs, while many endothelial and apoptosis pathway genes were increased relative to

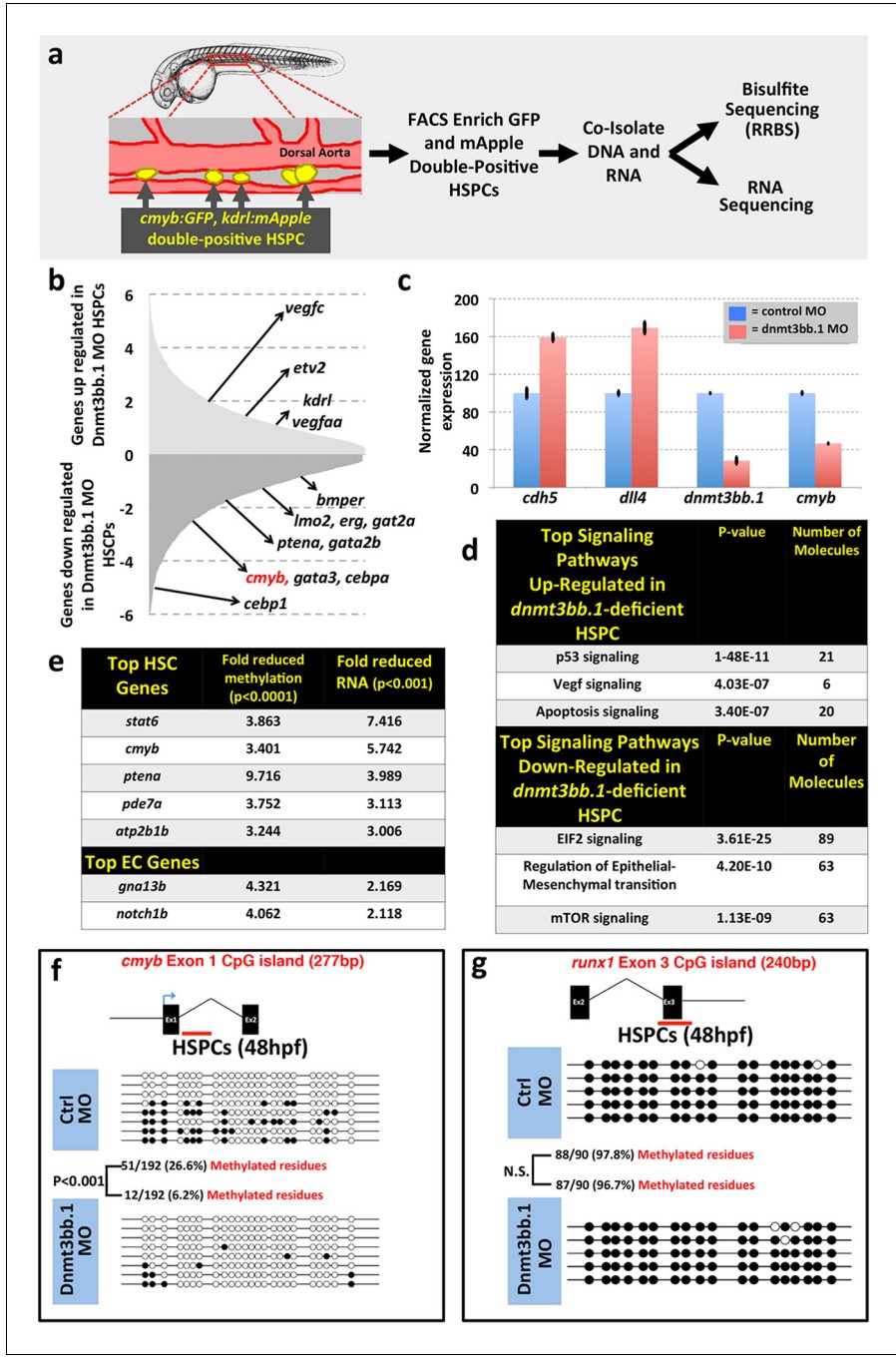

**Figure 3.** Dnmt3bb.1is necessary for epigenetic regulation of hematopoietic gene expression. (**a**) Schematic diagram showing the experimental procedure for the isolation of GFP/mApple double-positive HSPCs from *cmyb: GFP, kdrl:mApple* double transgenic zebrafish and use of DNA and RNA from these HSPC for next-gen sequencing projects (**b**) RNAseq analysis showing differentially expressed endothelial and hematopoietic genes in dnmt3bb.1 deficient HSPCs. (**c**) qRT-PCR analysis of selected genes from the RNAseq analysis. (**d**) Ingenuity Pathway Analysis (IPA) of the RNAseq data showing top pathways up- or down-regulated in dnmt3bb.1-deficient HSPCs. (**e**) Top five HSC-expressed genes with the most significantly reduced DNA methylation and expression from bisulfite and RNAseq analysis, respectively, as well as the top 2 endothelial genes. (**f,g**) Bisulfite sequencing analysis of DNA methylation at the *cmyb* intron 1 (f) and *runx1* exon 3 (g) CpG islands in DNA isolated from control ('Ctrl MO', top) and dnmt3bb.1 ('Dnmt3bb.1 MO', bottom) morpholino-injected HSPCs, showing strongly reduced methylation of the *cmyb* CpG island but not the *runx1* CpG island in dnmt3bb.1-deficient animals.

*Figure 3 continued on next page*

*Figure 3 continued*

The following source data and figure supplement are available for figure 3:

**Source data 1.** Hematopoieticand Vascular genes with reduced methylation and altered expression in dnmt3bb. 1 morpholino-injected animals.

**Figure supplement 1.** Analysisof RNA seq and DNA methylation.

control HSPCs (*Figure 3b,c*, *Figure 3—figure supplement 1b*). Ingenuity pathway analysis (www.qiagen.com/ingenuity) confirmed elevated VEGF/angiogenesis and apoptosis pathway gene expression indicating lack of endothelial to hematopoietic transition and induction of apoptosis in dnmt3bb.1-deficient HSPCs (*Figure 3d*). However, RRBS analysis revealed that very few of the hematopoietic or endothelial genes with significantly altered expression displayed significantly decreased DNA methylation (*Figure 3—source data 1*). *cmyb* was one of the most affected and highly correlated of the hematopoietic genes that showed both significantly reduced expression in dnmt3bb.1-deficient HSPCs and significantly reduced methylation of an adjacent promoter or gene body CpG island (*Figure 3e*, *Figure 3—source data 1*). Direct sequencing of PCR-amplified fragments from bisulfite treated DNA confirmed reduced methylation in the *cmyb* intron 1 CpG island in dnmt3bb.1-deficient HSPCs (*Figure 3f*), but no change in the *runx1* exon 3 CpG island (*Figure 3g*), or in CpG islands from a number of other hematopoietic and endothelial genes showing significantly altered expression in dnmt3bb.1-deficient HSPC (*Figure 3—figure supplement 1c,d*). These results suggest that *cmyb* is a target of dnmt3bb.1-mediated regulation in HSPCs.

To further validate our findings regarding methylation of hematopoietic genes using an independent method, we identified 319 genes that were annotated as being expressed during definitive hematopoiesis from a comprehensive database of zebrafish gene expression patterns (http://zfin.org). 57 of these genes were found to have a total of 72 CpG islands in their gene bodies (we concentrated on gene body CpG islands since these are associated with active gene expression [*Challen et al., 2012*; *Wu et al., 2010*; *Trowbridge and Orkin, 2012*; *Gao et al., 2011*]). We carried out bisulfite restriction analysis to determine whether these 72 CpG islands are methylated in blood cells, using genomic DNA isolated from circulating blood cells obtained after amputating the tails of 5–7-day-old zebrafish (*Figure 4a*), and determined that 10 of the 72 CpG islands are methylated in blood cells. To determine whether methylation of these CpG islands was dnmt3bb.1-dependent, we carried out bisulfite sequencing on each of the 10 methylated CpG islands using 5 dpf blood cell genomic DNA isolated from either control or dnmt3bb.1 morpholino-injected zebrafish. Only one CpG island showed dnmt3bb.1-dependent changes in methylation, the *cmyb* intron 1 CpG island (*Figure 4b*). No change was detected in the methylation of the *runx1* exon 3 CpG island (*Figure 4c*), or in any of the other eight hematopoietic genes containing methylated CpG islands in blood cells (*Figure 4d–k*). These results support the findings of our RRBS analysis above, suggesting that *cmyb* is a key proximal hematopoietic target of dnmt3bb.1.

## Dnmt3bb.1 is sufficient to drive hematopoietic gene expression

To examine whether Dnmt3bb.1 can promote *cmyb* gene expression in endothelial cells, we drove mosaic, pan-endothelial expression of a Dnmt3bb.1-GFP fusion protein by injecting a *Tol2(kdrl:dnmt3bb.1-gfp)* transgene (*Figure 5a* and *Figure 5—source data 1*) into zebrafish embryos, and assayed for expression of *cmyb* in Dnmt3bb.1-GFP positive cells. In wild type animals, *cmyb* is expressed in a limited number of endothelial precursors found in the ventral floor of the dorsal aorta in the trunk. In *Tol2(kdrl:dnmt3bb.1-gfp)* transgene-injected animals Dnmt3bb.1-GFP-positive cells were consistently also *cmyb* positive, and double–positive cells were found throughout the vasculature, not only in their normal location in the trunk (*Figure 5b,c,d*), but also in ectopic locations such as the cranial vasculature (*Figure 5b,e,f*). We observed similar induction of *cmyb* expression in *dnmt3bb.1-gfp* expressing cells when the transgene was injected into *runx1* mutant animals lacking endogenous cmyb expression in HSPCs (*Figure 5b,g,h*), showing that the forced expression of dnmt3bb.1 can promote cmyb expression in endothelial cells even in the absence of runx1 function.

Since endothelial cells might have intrinsic preconditioning making them more amenable to differentiating into HSPCs, we examined whether cmyb and other hematopoietic genes could also be

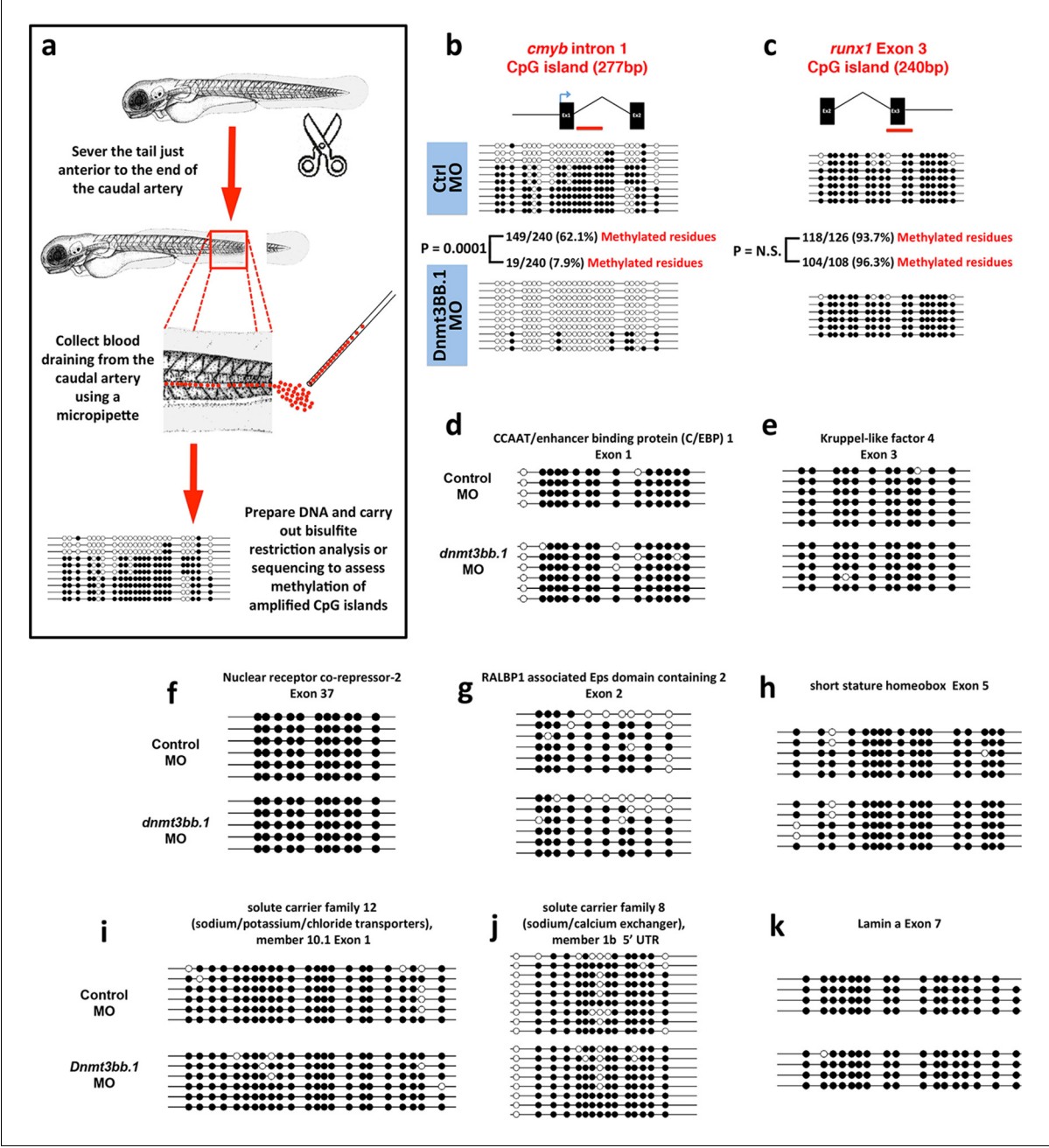

**Figure 4.** Measurement of DNA methylation by bisulfite sequencing of gene body CpG islands from all known hematopoietic genes. (a) Schematic diagram showing the method used to collect blood from 5–7 dpf embryos for bisulfite analysis. (b–k) Bisulfite sequencing of blood cell genomic DNA from 5–7 dpf control (top) and dnmt3bb.1 (bottom) morpholino-injected animals. The results are shown for ten different sequences identified in a genome-wide screen for gene body CpG islands methylated in blood cells: cmyb (b), runx1 (c), CCAT/enhancer binding protein 1-Exon 1 (d), Kruppel like factor 4-Exon 3 (e), Nuclear receptor co-repressor-2-Exon 37 (f), RALBP1 associated domain containing 2-Exon 2 (g), short stature homeobox-Exon 5 (h), solute carrier family 12-Exon 1 (i) and solute carrier family 8–5' UTR (j), lamin a-Exon 7 (k). Open circles represent unmethylated and filled circles represent methylated cytosine residues from CpG dinucleotides.

induced in non-endothelial cells by forced overexpression of *dnmt3bb.1*. We mosaically mis-expressed Dnmt3bb.1-GFP fusion protein in cells in the early zebrafish embryo, well before the initiation of normal hematopoietic development, by heat-shocking *Tol2(hsp70:dnmt3bb.1-gfp)* transgene-injected animals at the blastula stage (3 hpf) and then assaying expression of *cmyb* and downstream hematopoietic lineage markers at the shield stage (6 hpf) by WISH and qRT-PCR

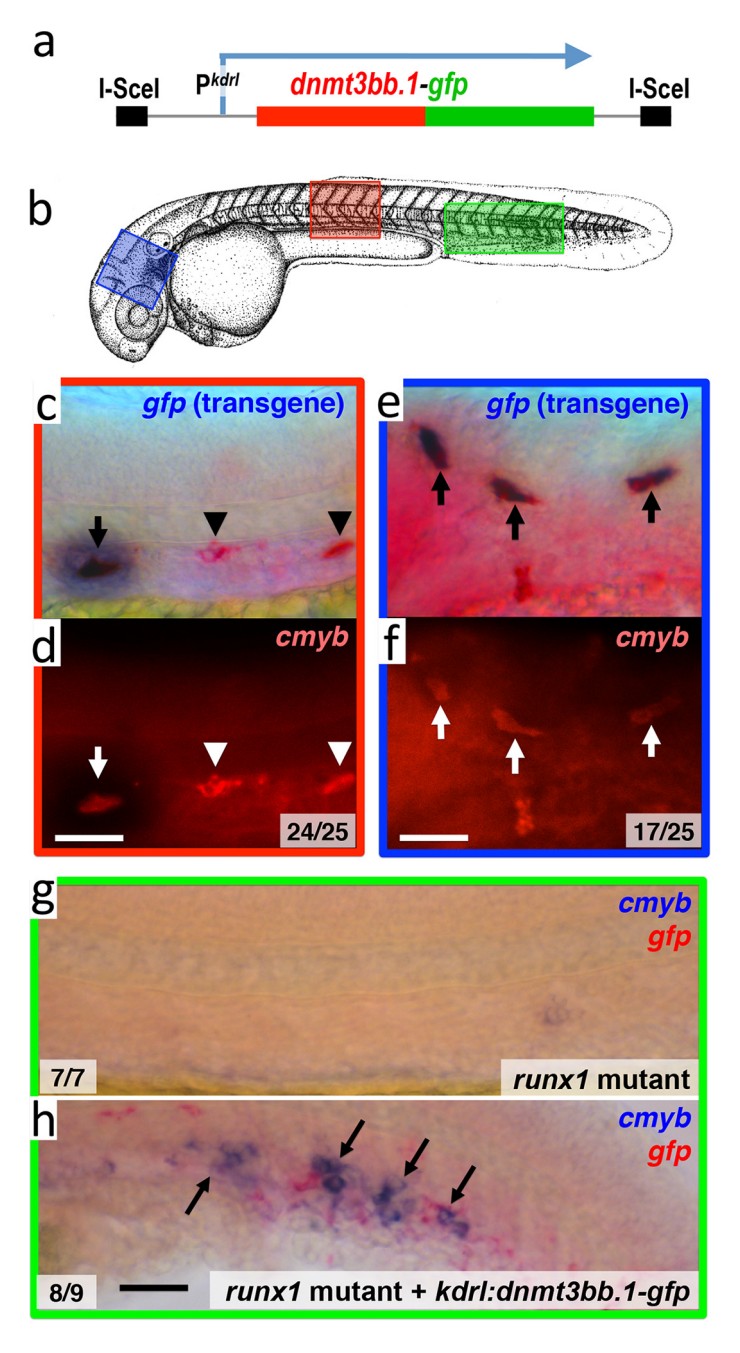

**Figure 5.** Dnmt3bb.1 is sufficient for *cmyb* gene expression in the endothelium. (**a**) Schematic diagram of the *I-Sce1(kdrl:dnmt3bb.1-gfp)* transgene used for the pan-endothelial expression of a *dnmt3bb.1-gfp* fusion protein. (**b**) Camera lucida drawing of a 24 hpf zebrafish embryo with red, blue and green boxes noting the approximate regions of the head, trunk and trunk/tail shown in in situ hybridization images in panels c-h. (**c,e**) Double whole-mount in situ hybridization of the head (e) and trunk (c) of 36 hpf *I-Sce1(kdrl:dnmt3bb.1-gfp)* transgene-injected zebrafish probed for *gfp* (blue) and *cmyb* (red). (**d,f**) fluorescence images of in situ hybridization corresponding to panels c and e, respectively. In panels c-e, endothelial cells expressing both the *gfp* transgene and *cmyb* are noted with arrows, while normal trunk HSC expressing only *cmyb* but not *gfp* are noted with arrowheads. (**g, h**) Double whole-mount in situ hybridization of the trunk/tail of a 36 hpf *runx1* mutant (g) and a 36 hpf *runx1* mutant injected with *I-Sce1(kdrl:dnmt3bb.1-gfp)* transgene (h), probed for *cmyb* (blue) and *gfp* (red). The *cmyb* gene is not expressed in *runx1* mutants, but injection of *dnmt3bb.1-egfp* fusion protein into *runx1* mutants results in the appearance of *cmyb/gfp* double-positive cells. Scale bars 25 μm in e,f and 100 μm in h.

*Figure 5 continued on next page*

*Figure 5 continued*

The following source data is available for figure 5:

**Source data 1.** Mosaic expression of dnmt3bb.1:gfp in endothelial cells induces cmyb expression.

(*Figure 6a,b*). As expected, heat-shocked control *Tol2(hsp70:gfp)* transgene-injected animals do not express *cmyb, rag1*, or *l-plastin* (*Figure 6c-e,k*). In contrast, heat shocked animals expressing Dnmt3bb.1-GFP fusion protein from the injected *Tol2(hsp70:dnmt3bb.1-gfp)* transgene showed robust mosaic expression of *cmyb, rag1*, and *l-plastin* (*Figure 6f-h,k*). Dnmt3bb.1-GFP-injected embryos also express lymphoid markers *ikaros* and *rag2*, as well as *hemoglobin ae1 (hbae1)*, a marker of both primitive and definitive erythroid cells (*Jin et al., 2009),* showing induction of erythroid lineage genes also occurs (*Figure 6—figure supplement 1a–h*), but they do not express endothelial markers *cadherin 5 (cdh5)* and *ets-related protein (etsrp,* also known as *etv2)* (*Figure 6—figure supplement 1i–l*). This suggests that dnmt3bb.1 specifically promotes hematopoietic development without inducing mesoderm, endothelial precursors, or 'hemangioblasts'.

To further investigate the ability of dnmt3bb.1 to promote functional properties of HSPC in naive blastula cells, we examined whether cells from *Tol2(hsp70:dnmt3bb.1-gfp)*-injected donors would home to the thymus when transplanted into host larvae. We dissociated 5 hpf heat shocked WT, *Tol2(hsp70:dnmt3bb.1-gfp)*- or control *Tol2(hsp70:gfp)*-injected blastula embryos and transplanted the dissociated cells into the circulation of 48 hpf *Tg(cmyb:gfp)* (*North et al., 2007*) or *Tg(lck:egfp)* transgenic (*Langenau et al., 2004*) hosts by intravenous injection (*Figure 6l*). On day 5, 17/30 (57%) of the host animals that received cells from heat shocked *Tol2(hsp70:dnmt3bb.1-gfp)* donors showed donor cells (red fluorescent, rhodamine/biotin-dextran-labeled) homing to the thymus, as compared to only 11/38 (29%) of host animals that received cells from control heat -shocked *Tol2(hsp70:gfp)* donors (*Figure 6n–q*). To be conservative, we decided to include counts of cell numbers in the thymus and heart/aortic arches only for the animals that showed transplanted cells in the thymus (*Figure 6q*). The thymus-positive *dnmt3bb.1-gfp* transplanted hosts had approximately twice as many cells in the thymus compared to the thymus-positive *gfp* control transplanted hosts, while the number of intravenously injected cells that were lodged non-specifically in the heart and aortic arches was comparable.

## Cmyb is a target of dnmt3bb.1 regulation

We carried out additional experiments to further examine whether cmyb is a proximal target of dnmt3bb.1. Bisulfite sequencing reveals that the *cmyb* intron 1 CpG island becomes methylated in DNA from heat -shocked mosaic *Tol2(hsp70:dnmt3bb.1-gfp)* transgene-injected shield stage animals, but not in DNA from heat shocked control *Tol2(hsp70:gfp)* transgene-injected animals (*Figure 7a*). A separate *cmyb* 5' CpG island that is not normally methylated in HSPC in wild type animals does not become ectopically methylated in heat-shocked *Tol2(hsp70:dnmt3bb.1-gfp)* transgene-injected shield stage animals (*Figure 7b*). Chromatin immunoprecipitation (ChIP) of DNA from heat-shocked *Tol2 (hsp70:gfp)* or *Tol2(hsp70:dnmt3bb.1-gfp)* transgene-injected or *Tg(hsp70:tcf-gfp)* germline transgenic (*Lewis et al., 2004*) shield-stage animals (*Figure 7c*) using anti-GFP antibodies shows that dnmt3bb.1-GFP (but not Tcf-GFP) binds to two independent DNA sequences from the *cmyb* intron 1 CpG island (*Figure 7d*, rows 1 and 2). Dnmt3bb.1-GFP does not bind to non-CpG island *cmyb* DNA from intron 1 or exon 5, or to CpG island DNA from the *runx1* or *ntla* gene loci (*Figure 7d*, rows 3–6). As an additional ChIP control, Tcf-GFP (but not Dnmt3bb.1-gfp) shows specific association with *sox3* DNA sequences containing known TCF binding sites (*Lee et al., 2006*) (*Figure 7d*, row 7), but does not associate with any dnmt3bb.1 DNA sequence. ChIP using anti-methyl cytosine (*Figure 7e*) confirms methylation of the *cmyb* intron 1 CpG island in animals expressing dnmt3bb.1-GFP but not GFP alone (*Figure 7e*, rows 1 and 2), as well as 'constitutive' methylation of the *runx1* and *ntla* CpG islands (*Figure 7e*, rows 5 and 6). Together, these results suggest that Dnmt3bb.1 methylates the *cmyb* intron 1 CpG island to maintain the expression of *cmyb* in HSPCs, and that loss of Dnmt3bb.1 results in failure to maintain *cmyb* expression and loss of HSPCs.

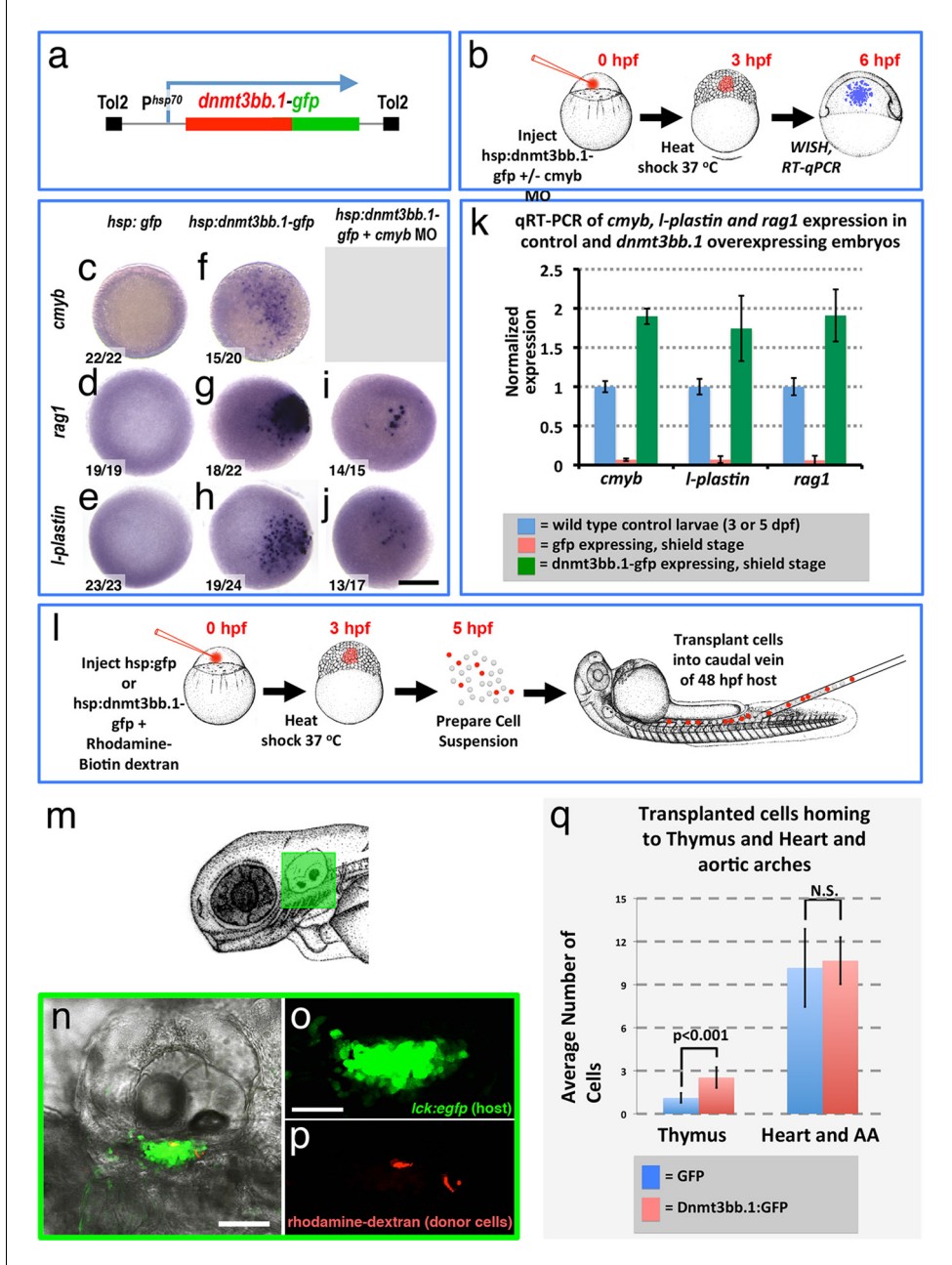

**Figure 6.** Dnmt3bb.1 is sufficient for HSPC and downstream lineage gene expression. (**a**) Schematic diagram of the *Tol2(hsp70:dnmt3bb.1-gfp)* transgene used for ubiquitous heat shock-inducible expression of *dnmt3bb.1-gfp* fusion protein. (**b**) Camera lucida drawing showing the experimental protocol employed for early embryonic induction of dnmt3bb.1. One cell-stage embryos are injected with *Tol2(hsp70:dnmt3bb.1-gfp)* transgene +/- cmyb morpholino (MO), raised to 3 hpf (mid-blastula stage), heat shocked, allowed to further develop to 6 hpf (shield stage), then either fixed for whole mount in situ hybridization (WISH) or collected to prepare RNA for RT-qPCR. (**c–j**) Whole mount in situ hybridization of 6 hpf control *Tol2(hsp70:gfp)* transgene-injected (c-e), *Tol2(hsp70:dnmt3bb.1-gfp)* transgene-injected (f-j), or *Tol2(hsp70:dnmt3bb.1-gfp)* transgene plus *cmyb* morpholino (MO)-injected (i,j) zebrafish embryos, probed for *cmyb* (c,f), *rag1* (d,g,i), or *l-plastin* (e,h,j). (**k**) Quantitative RT-PCR analysis of *cmyb, l-plastin,* and *rag1* transcript levels in positive control untreated 3 dpf (for *cmyb* and *l-plastin*) or 5 dpf (for *rag1*) larvae (blue columns), negative control shield stage (6 hpf) embryos injected with a *Tol2(hsp70:egfp)* transgene and heat shocked at 3 hpf (red columns), or shield stage (6 hpf) embryos injected with an *Tol2(hsp70:dnmt3bb.1-egfp)* transgene and heat shocked at 3 hpf (green columns). (**l**) Schematic drawing showing the experimental protocol employed for early embryonic induction of dnmt3bb.1-gfp and subsequent transplantation
*Figure 6 continued on next page*

*Figure 6 continued*

of cells into the circulation. One cell-stage embryos were injected with *Tol2(hsp70:egfp)* or *Tol2(hsp70:dnmt3bb.1-egfp)* transgenes, raised to 3 hpf (mid-blastula stage), heat shocked, and allowed to develop to 5 hpf, at which point the embryos were dissociated and cell suspensions was prepared. Dissociated cells were injected into the circulation of 48 hpf host larvae using a borosilicate needle without filament. (m) Camera lucida drawing of a 72 hpf zebrafish embryo with a green box noting the approximate regions containing the thymus. (n–p) Confocal images of a 5 dpf *Tg(lck:GFP)*cz1 host animal showing transplanted cells colonizing the thymus. Transmitted light image (n), green fluorescent lck:gfp positive host thymus (n,o), and red fluorescent rhodamine-dextran positive donor cells populating the thymus (p). (q) Quantitation of the number of control GFP and dnmt3bb.1-GFP expressing cells in the thymus and heart/aortic arch region. Images in panels c-f,r,s are dorsal views and panels n-p are lateral views. Scale bars = 150 μm in c-j, 25 μm in n, 10 μm in o.

The following figure supplement is available for figure 6:

**Figure supplement 1.** Dnmt3bb.1induces hematopoietic but not endothelial gene expression in early embryos.

## dnmt3bb.1 promotes hematopoietic cell fate via cmyb

Previous studies have shown that Cmyb function is necessary and sufficient for the development of definitive blood cell lineages and for the expression of downstream lineage genes (*Soza-Ried et al., 2010*; *Zhang et al., 2011*; *Mukouyama et al., 1999*). In zebrafish, a small proportion of animals homozygous null for either *cmyb* or *runx1* survive to adulthood. These 'escapers' show severe reduction in blood cell number in circulation and in the anterior kidney (a blood-forming organ in adult zebrafish). In a similar fashion, we find that 10–12% of homozygous *dnmt3bb.1* mutants also survive to adulthood (*Figure 8a,b*), and the anterior kidneys in these surviving mutants also exhibit severely reduced hematopoietic cellularity (*Figure 8c–f*). The circulating erythrocytes that are present in adult mutants are mostly malformed (*Figure 8g–i*). Interestingly, sorted HSPCs from dnmt3bb.1-deficient 48 hpf embryos also display malformations including abnormally shaped, bi-lobed or even multiple distinct nuclei (*Figure 8j,k*). Again, these results are similar to findings with *cmyb* mutants (*Soza-Ried et al., 2010*), supporting the idea that Dnmt3bb.1 exerts its effects on HSPC fate maintenance and hematopoietic gene expression via Cmyb.

To examine whether ectopic Dnmt3bb.1 and Cmyb expression in the early embryo results in similar induction of downstream hematopoietic gene expression, we drove *cmyb* expression in the blastula by the injection of a *Tol2(hsp70:cmyb-2A-mCherry)* transgene (*Figure 8—figure supplement 1a*). Like overexpression of dnmt3bb.1-gfp in early embryos, heat shock induction of embryos injected with *Tol2(hsp70:cmyb-2A-mCherry)* leads to ectopic expression of *hbae1, rag1, l-plastin, ikaros, lck* and *rag2* (*Figure 8—figure supplement 1b–m*), but not endothelial markers *cdh5* or *etsrp* (*Figure 8—figure supplement 1n–q*).

To more directly examine whether *cmyb* is required downstream from *dnmt3bb.1* for induction of hematopoietic lineage genes, we co-injected the *Tol2(hsp70:dnmt3bb.1-gfp)* transgene together with a previously published *cmyb* morpholino(*Grabher et al., 2011*) and heat shocked as described above. We saw dramatic reduction in *l-plastin* and *rag1* in dnmt3bb.1-GFP over-expressing embryos co-injected with *cmyb* morpholino compared to embryos injected with *Tol2(hsp70:dnmt3bb.1-gfp)* transgene alone (*Figure 6g,h,i,j*).

Together, these results confirm that cmyb function is both necessary and sufficient for the expression of hematopoietic lineage genes, and further shows that the expression of hematopoietic genes cannot be induced by dnmt3bb.1 in the absence of cmyb. This suggests that *cmyb* is a key locus regulated by *dnmt3bb.1* during hematopoietic differentiation.

## Discussion

In this manuscript, we show that an epigenetic regulatory factor promotes the maintenance of hemtopoietic cell fate downstream from an established genetic pathway required for the specification of HSPCs The *de novo* DNA methyltransferase *dnmt3bb.1,* the closest zebrafish ortholog of the mammalian DNA methyltransferase DNMT3B, is expressed specifically in cmyb-positive HSPC emerging from hemogenic endotheium in the ventral floor of the dorsal aorta. HSPC specification is controlled by a genetically programmed Notch-Runx1-Cmyb pathway (*Burns et al., 2005*; *Gering and Patient,*

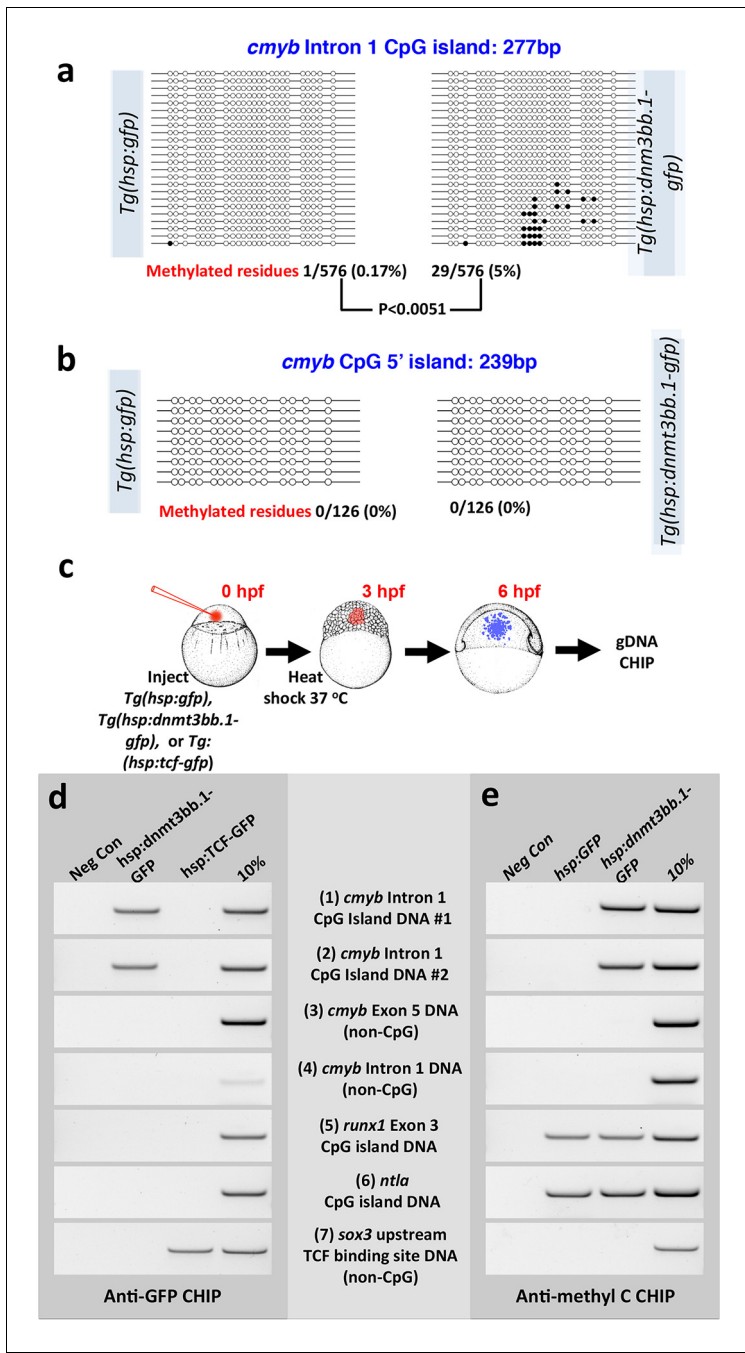

**Figure 7.** Dnmt3bb.1 induces methylation of *cmyb* exon 1 CpG island DNA in early embryos. (**a,b**) Bisulfite sequencing of genomic DNA from 6 hpf control *Tol2(hsp70:gfp)* transgene- or *Tol2(hsp70:dnmt3bb.1-gfp)* transgene-injected embryos to assess methylation of the cmyb intron 1 CpG island (a) or cmyb 5' CpG island (b). Open circles represent unmethylated cytosine residues and filled circles represent methylated cytosine residues from CpG dinucleotides. Methylation of the intron 1 CpG island is detected in dnmt3bb.1-gfp expressing embryos, but not in control gfp-expressing embryos. (**c–e**) Chromatin immunoprecipitation (CHIP) using anti-GFP or anti-methylcytosine antibodies to detect Dnmt3bb.a-GFP (or TCF-GFP control) fusion protein binding or cytosine methylation in CpG islands (CGI), respectively. (**c**) Schematic diagram showing the experimental procedure used for CHIP analysis, (**d**) Anti-GFP CHIP with either no added chromatin (negative control; column 1) or with chromatin from either hsp:dnmt3bb.1-GFP injected embryos (column 2) or from hsp:TCF-GFP injected embryos (column 3; CHIP positive control). Column 4 is 10% of input chromatin from hsp:dnmt3bb.1-GFP injected embryos. (**e**) Anti-methylcytosine CHIP with either no added chromatin (negative control; column 1) or using chromatin from hsp:GFP injected embryos (column 2; negative control for dnmt3bb.1-gfp methylation) or from

*Figure 7 continued on next page*

*Figure 7 continued*

hsp:dnmt3bb.1-GFP -injected embryos (column 3). Column 4 is 10% of input chromatin from hsp:dnmt3bb.1-GFP injected embryos. Primers used for the CHIP experiments in both e and f were: (1) cmyb Intron 1 CpG Island DNA, site #1, (2) cmyb Intron 1 CpG Island DNA, site #2, (3) cmyb Exon 5 DNA (non-CpG island), (4) cmyb Intron 1 DNA (non-CpG island), (5) runx1 Exon 3 CpG island DNA, (6) ntla CpG island DNA, and (7) sox3 upstream TCF binding site DNA (non-CpG).

---

*2005*; *Bresciani et al., 2014*; *Duncan et al., 2005*; *Bertrand et al., 2010b*; *Kumano et al., 2003*). Runx1 mutants and runx1 morpholino-injected zebrafish display reduced expression of *dnmt3bb.1*, as do Notch-deficient *mind bomb* mutants. Since *runx1* and *mind bomb* mutant embryos have strongly reduced numbers of HSPCs, reduced dnmt3bb.1 expression may reflect reduced HSPC numbers rather than the loss of direct gene regulation. Conversely, overexpression of runx1 leads to increased expression of *dnmt3bb.1* in hemogenic endothelium. Since mouse *Dnmt3b* is also expressed by developing CD34$^+$ HSPCs in the endothelium of early mouse embryos (*Watanabe et al., 2004*), and is up-regulated by Oct4 in direct conversion of fibroblasts into multi-potent blood progenitors in cell culture (*Szabo et al., 2010*), this suggests that the role of these orthologous DNMT proteins may be conserved during early hematopoietic development in vertebrates.

Our loss- and gain-of-function data show that *dnmt3bb.1* is required for HSPC maintenance. In dnmt3bb.1 mutants or dnmt3bb.1 morpholino-injected animals HSPC are initially specified but sub-sequently undergo apoptosis with mutant embryos showing gradual reduction in cmyb+ HSPCs. Mutants and morphants also show later defects in downstream blood lineages, as evidenced by the reduced expression of specific markers of the myeloid, lymphoid, and erythroid lineages. These loss-of-function studies show that dnmt3bb.1 is necessary for HSPC induction; additional gain-of–function experiments we performed show that dnmt3bb.1 is also sufficient to drive hematopoietic development. Ectopic mosaic overexpression of Dnmt3bb.1 activates hematopoietic gene expression in non-hemogenic endothelium, and even in 'naive' cells in the early blastula well before normal emergence of HSPC or specification of any blood cell lineages. Previous studies have shown that *cd41:gfp* transgene-positive HSPC isolated from transgenic zebrafish larvae populate the thymus upon transplantation (*Bertrand et al., 2008*). Transplanted cells from dnmt3bb.1-overexpressing blastulae also preferentially 'home' to the thymus and to the kidney (data not shown), suggesting that these dnmt3bb.1-overexpressing cells are recapitulating at least one functional property of HSPCs. The remarkable capacity of dnmt3bb.1 to activate hematopoietic gene expression in non-hematopoietic cells appears to reflect a fairly direct conversion towards the hematopoietic lineage rather than a more general induction of mesoderm or hemogenic endothelium, since we do not observe ectopic expression of endothelial markers in dnmt3bb.1-overexpressing blastulae, nor do we detect ectopic expression of other markers of mesoderm (unpublished results). Another epige-netic factor, the histone demethylase Kdm2b, was recently shown to be able to promote the conversion of fibroblasts into iPS cells on its own (*Liang et al., 2012*), reinforcing the idea that epigenetic factors can facilitate reprogramming independent of key specification factors (such as Oct4, Klf2 and Sox2 during iPS conversion). Further investigation into the capacity of dnmt3bb.1 and its closest mammalian ortholog DNMT3B to promote the conversion of somatic cells into blood cells, and the potential usefulness of this for generation of blood cells from non-hematopoietic progeni-tors in vitro, will be of great interest.

Our results strongly suggest that Dnmt3bb.1 exerts its hematopoietic effects via Cmyb. Com-bined whole-genome analysis of gene expression and DNA methylation in HSPC isolated from dou-ble-transgenic zebrafish embryos by flow cytometry shows that *cmyb* is one of the most significant genes with both reduced expression and reduced DNA methylation in dnmt3bb.1-deficient HSPC. Expression of number of hematopoietic genes is increased or decreased in dnmt3bb.1-deficient HSPCs, but only a handful of these genes display significantly reduced methylation in dnmt3bb.1-deficient animals and cmyb is one of the most highly significant of these. We do not detect signifi-cant changes in DNA methylation in *runx1, gata2, scl, ikaros, lck, rag1, l-plastin*, or *pu.1*, suggesting that other HSPC and hematopoietic lineage genes are not direct targets of dnmt3bb.1. The impor-tance of Cmyb downstream from Dnmt3bb.1 was further confirmed by an unbiased DNA

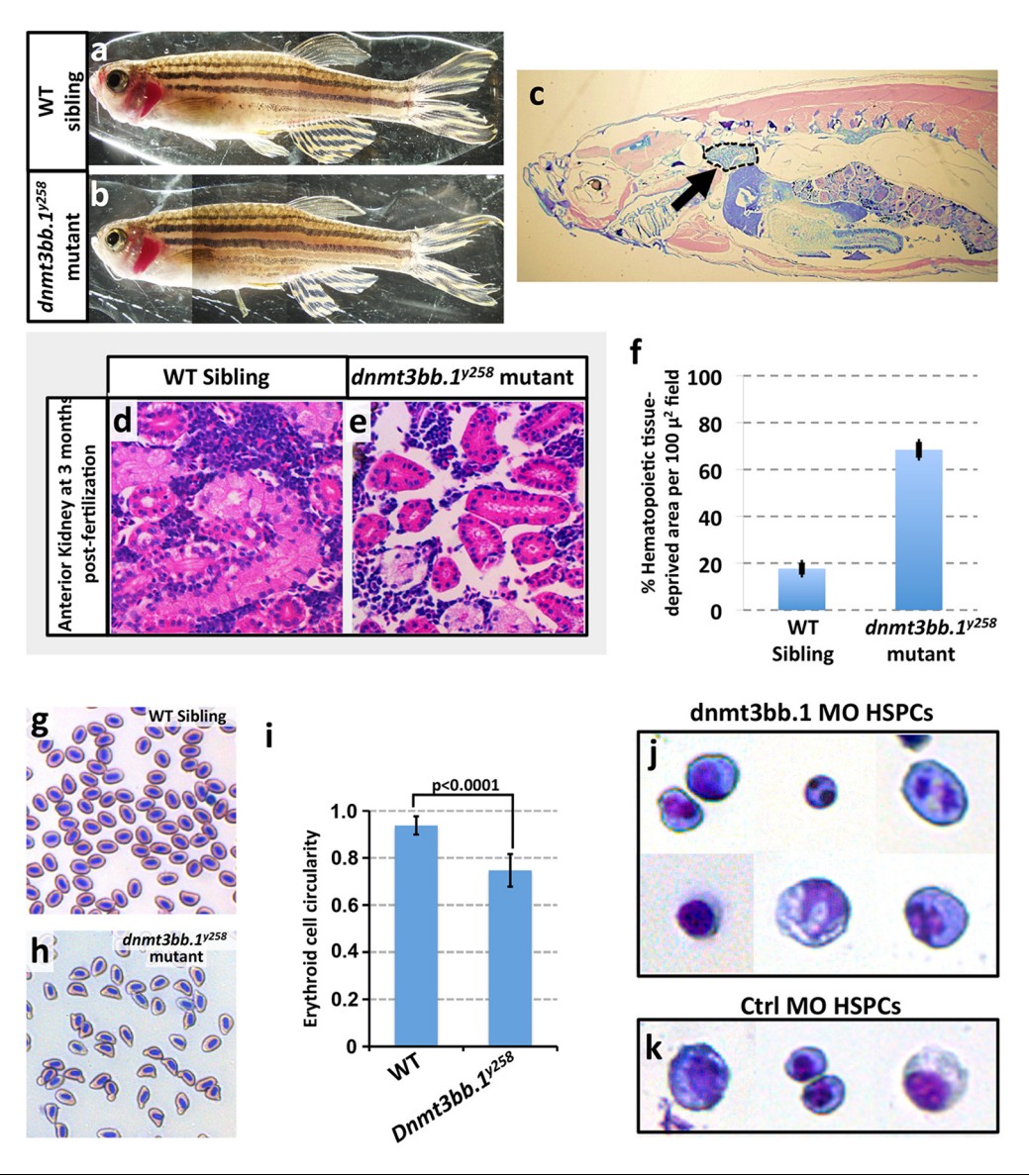

**Figure 8.** Hematopoieticdefects in adult *dnmt3bb.1*[y258] Mutants. (**a,b**) Wild type sibling (a) and *dnmt3bb.1*[y258]homozygous mutant (b) adult zebrafish. (**c**) Representative low-magnification sagittal section through a *dnmt3bb.1*[y258] homozygous mutant adult zebrafish, showing the head kidney (marked an arrow and surrounded by a dashed line), a major site of hematopoiesis in adult zebrafish. (**d,e**) Representative higher magnification images of sagittal sections through the head kidneys of wild type sibling (d) and *dnmt3bb.1*[y258] mutant (e) adult zebrafish, stained with hematoxylin/eosin. (**f**) quantitation of the percentage of the hematopoietic (non-kidney tubule) area that is acellular in head kidney sections from wild type sibling (left column) and *dnmt3bb.1*[y258] mutant (right column) adult zebrafish. (**g,h**) Giemsa -stained blood smears from wild type sibling (g) and *dnmt3bb.1*[y258] mutant (h) adult zebrafish. (**i**) quantitation of the adult erythroid cell circularity in WT and *dnmt3bb.1*[y258] mutant (see Materials and methods), measured from blood smears such as those in panels g and h. (**j,k**) Giemsa stained FACS-sorted HSPC (as in *Figure 3a*) from 36 hpf dnmt3bb.1 (j) or control (k) morpholino-injected zebrafish.

The following figure supplement is available for figure 8:

**Figure supplement 1.** Cmybinduces hematopoietic but not endothelial gene expression in early embryos.

methylation survey of CpG islands associated with 319 genes annotated as 'hematopoietic' in a zebrafish gene expression database (ZFIN). Only 10 of these genes have methylated gene body CpG islands in blood cells, and of these ten only the cmyb gene shows reduced methylation (of its intron 1 CpG island) in dnmt3bb.1-deficient animals. We were able to validate that the intron 1 CpG island is indeed a *bona fide* target of dnmt3bb.1 binding and DNA methylation using chromatin immunoprecipitation (ChIP). ChIP with an anti-GFP antibody shows dnmt3bb.1-GFP binds specifically to the *cmyb* intron 1 CpG island, and an anti-Methyl-C antibody confirms specific methylation of this CpG island by dnmt3bb.1-GFP. The mechanisms targeting DNA methyltransferases to some but not other CpG islands is still poorly understood. Class 3 *de novo* DNMTs are known to interact with a number of different transcription factors in vitro (*Brenner et al., 2005*; *Fuks et al., 2001*; *Suzuki et al., 2006*), and these interactions are thought to influence their genomic targeting specificity (*Jurkowska et al., 2011*). However, these interactions and their functional consequences on gene expression have not been validated in vivo (*Hervouet et al., 2009*). Besides transcription factors, DNMTs are also known to interact with other epigenetic regulators, mainly histone- modifying enzymes. In zebrafish, dnmt3bb.2 (previously known as Dnmt3) has been shown to interact with H3K9 methyltransferase G9a and regulate *lef1* expression (*Rai et al., 2010*). It is possible that a variety of genetic and epigenetic factors may cooperate with dnmt3bb.1 to facilitate its interaction with the cmyb locus.

Our functional data also strongly implicate cmyb as a key target regulated by dnmt3bb.1 during hematopoietic development. Loss of Dnmt3bb.1 results in phenotypes that closely mirror those resulting from loss of cmyb. Like previously reported zebrafish cmyb mutants (*Soza-Ried et al., 2010*), a small proportion (10–15%) of dnmt3bb.1 mutants survive past 6–9 weeks. Although these dnmt3bb.1 mutant 'escapers' appear externally normal, they have hypoplasia of the anterior kidney marrow and abnormal red blood cell morphology – again, phenotypes extremely similar to those seen in surviving adult cmyb mutants (*Soza-Ried et al., 2010*). In addition, overexpression of cmyb in the early blastula results in induction of hematopoietic gene expression virtually indistinguishable from that seen upon overexpression of dnmt3bb.1. Finally, dnmt3bb.1-induced expression of hematopoietic lineage genes in blastulae can be prevented by knock-down of cmyb, showing that cmyb is necessary for dnmt3bb.1-mediated hematopoietic gene expression. Taken together, our results strongly suggest that dnmt3bb.1 promotes continued hematopoietic development by methylating *cmyb* gene body DNA and maintaining activation of the cmyb locus. This provides an explanation for how transient expression of the HSPC-specific Runx1 transcription factor, which is required for HSPC specification during early development, can lead to the continued expression of cmyb required for prolonged maintenance of hematopoietic cell fate (*Figure 9*). Runx1 is required to activate expression of cmyb in HSPC, and it can do so in the absence of Dnmt3bb.1. However, Runx1 also activates the expression of Dnmt3bb.1, which 'marks' *cmyb* gene body DNA to promote

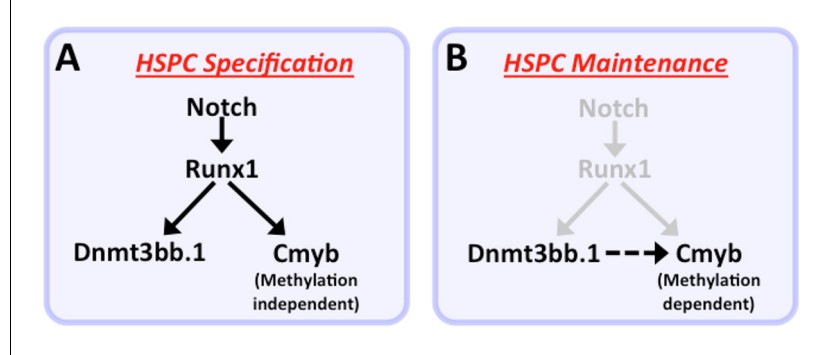

**Figure 9.** Model for regulation of HSPC cell fate during development. (a) Notch-Runx1 signaling controls specification of HSPCs in the ventral wall of the dorsal aorta during early embryogenesis. This signaling pathway initiates HSPC expression of both the key transcription factor Cmyb and the epigenetic regulator Dnmt3bb.1. (b) As development proceeds Runx1 expression is down-regulated and continued maintenance of active Cmyb expression in HSPC depends on Dnmt3bb.1–mediated Cmyb gene body DNA methylation, ensuring maintenance of HSPC cell fate during the Runx1-independent phase.

cmyb activation. As Runx1 expresion is down-regulated, these *cmyb* gene body DNA marks help to maintain expression of *cmyb* in the absence of Runx1.

There are previous precedents for epigenetic regulation targeting one or a small subset of genes to facilitate or reinforce a specific developmental cell fate decision. In avian embryos, expression of the related DNMT3A gene along the neural plate boundary promotes specification of neural crest by methylating CpG islands in the Sox2 and Sox3 promoter regions and repressing expression of these neural specification genes, acting as a 'molecular switch' promoting neural crest cell fate (*Hu et al., 2012*). In the case of zebrafish dnmt3bb.1, HSPC maintenance appears to be promoted by gene body, not promoter, methylation of cmyb, and cmyb expression is activated by DNA methylation, not repressed. Gene body methylation has been reported in many different organisms, and it is often associated with active gene expression (*Tweedie et al., 1997*; *Zemach et al., 2010*), which is thought to support by promoting proper RNA splicing and by helping to maintain chromatin structure for active gene expression (*Suzuki and Bird, 2008*; *Jones, 2012*). Interestingly, a recent study showed that DNMT3B, the mammalian ortholog of zebrafish dnmt3bb.1, preferentially binds to and methylates gene body DNA (but not promoter or enhancer DNA) of actively transcribed genes in mouse embryonic stem cells (*Baubec et al., 2015*), reinforcing the idea that this particular DNA methyltransferase preferentially promotes gene body DNA methylation and gene activation.

Previous studies have implicated DNA methyltransferases in adult hematopoiesis, with some studies reporting that DNA methylation promotes HSPC renewal and/or expansion (*Challen et al., 2012*; *Tadokoro et al., 2007*) and others reporting that it promotes lineage differentiation (*Broske et al., 2009*; *Trowbridge et al., 2009*). A recent zebrafish study reported that the maintenance DNA methyltransferase 1 facilitates HSPC formation via C/ebpa (*Liu et al., 2015*). DNA methylation is also strongly implicated in hematopoietic malignancies; indeed, inhibitors of DNA methylation are currently used therapeutically in this context (*Flotho et al., 2009*). Our results highlight an important new role for DNA methyltransferases during the earliest stages of HSPC emergence in the developing embryo. The molecular mechanisms involved in generation and maintenance of HSPC from hemogenic endothelium are still not well understood, and our findings provide evidence that DNA methylation-mediated epigenetic modifications downstream from Runx1 help maintain HSPC fate during definitive hematopoiesis. DNMT3B, the closest mouse ortholog of zebrafish dnmt3bb.1, is also expressed in developing HSPC during embryogenesis, suggesting the developmental roles of these proteins may be conserved (*Watanabe et al., 2004*). Another recent report suggests that histone and chromatin modifications may also help maintain HSPC cell fate during mouse embryogenesis (*Tober et al., 2013*). Together, these findings suggest that a number of different epigenetic processes may contribute to generating HSPC cell fate during embryogenesis. It will be interesting to further examine whether and how epigenetic regulators interact with various hematopoeitic transcription factors regulating HSPC and hematopoietic lineage-specific development.

## Materials and methods

### Zebrafish stocks and embryos

Zebrafish lines used in this study are *Tg(cmyb:GFP)^zf169^* (*North et al., 2007*), *Tg(fli1: eGFP)^y1^* (*Lawson and Weinstein, 2002*), *mib^ta52b^* (*Itoh et al., 2003*), *runx1^w84X^* (*Sood et al., 2010*), *Tg(lck:GFP)^cz1^* (*Langenau et al., 2004*), *Tg(kdrl:mApple)^y278^*, *dnmt3bb.1^y258^* (an allele carrying premature stop codon at amino acid 172), *dnmt3bb.1^b1237^* (an allele carrying premature stop codon at amino acid 42), and the EK wild type line. Generation and genotyping of *dnmt3bb.1^y258^* and *dnmt3bb.1^b1237^* are described below. Zebrafish were mated, staged (*Kimmel et al., 1995*) and raised (*Westerfield, 2000*) as previously described.

### Generation of *dnmt3bb.1* mutants

A pair of TALEN constructs targeting Exon 6 of *dnmt3bb.1* were designed using previously published methods (*Dahlem et al., 2012*). 100 pg of in vitro synthesized RNAs for each arm were injected at one cell stage to induce mutations at the target site. F1 lines were generated from F0 injected founders showing high efficiency of somatic mutations. Carriers were identified by PCR genotyping and subsequently sequenced to identify the mutations. An allele with an 11-nucleotide deletion resulting in a premature stop codon was identified (*dnmt3bb.1^y258^*) and used for further

analysis. For genotyping, PCR primers were designed spanning the TALEN target site and the resulting amplicons were subjected to SacI restriction enzyme digest. The SacI site is destroyed in the mutant allele due to the 11-nucleotide deletion. Genotyping of in situ stained embryos were done using the same PCR primers and conditions. For PCR, template genomic DNA was isolated using standard DNA extraction buffer (http://zfin.org/zf_info/zfbook/chapt9/9.3.html).

## DNA constructs

A full length *dnmt3bb.1* zebrafish clone was obtained from Open Biosystems. *Tol2(hsp:mCherry), Tol2(hsp:cmyb-2A-mCherry), Tol2(hsp:gfp), Tol2(hsp:dnmt3bb.1-gfp),* and *ISceI(kdrl:dnmt3bb.1-gfp)* transgene constructs were generated using Gateway technology (*Kwan et al., 2007*). *Dnmt3bb.1* and *cmyb* coding sequences were amplified using sequence specific primers with attB sites. The resulting PCR products were cloned into a pDONR vector (Thermo Fisher Scientific, Waltham, MA) to generate middle entry vectors using BP clonase enzyme (Thermo Fisher Scientific, Waltham, MA). Correct plasmids determined by sequencing were used to generate final destination clones using LR clonase enzyme (Thermo Fisher Scientific).

## Riboprobe synthesis, mRNA synthesis, and DNA, RNA, and morpholino microinjection

Antisense riboprobes were generated using Roche DIG and FITC labeling mix. Full length or EST clones were restriction digested and transcribed using appropriate enzymes. Synthetically capped *runx1* mRNA was synthesized using mMessage mMachine kit (Thermo Fisher Scientific) and injected as previously published (*Burns et al., 2005*). mRNA, DNA constructs and morpholinos were injected at onecell stage embryos using a pressure injector as described previously (*Gore et al., 2011*). All morpholino sequences used in this study are listed in methods *supplementary file 1A*.

## In situ hybridization and immunohistochemistry staining

In situ hybridization and immunohistochemistry were carried out as described (*Gore et al., 2011*). For double WISH, we followed modifications as described previously (*Lauter et al., 2011*). For detecting cell death, primary polyclonal anti-active Caspase 3 (C4748, Sigma-Aldrich, St. Louis, MO) and monoclonal anti-GFP antibodies were used at 1:500 and 1:200 dilutions, respectively. Primary antibodies were detected using anti-rabbit Alexa-543 and anti-mouse Alexa-488 labeled secondary antibodies at 1:400 dilution. After staining, embryos were cleared in Vectashield mounting media (Vector Laboratories, Burlingame, CA) and imaged using a Leica TCS-SP5 II confocal microscope.

## Flow cytometry, DNA and RNA isolation

Cell sorting was performed as described previously (*Bertrand et al., 2010a*). Briefly, double transgenic embryos derived from carriers of *Tg(cmyb:GFP);Tg(kdrl:mApple)* were injected with control or dnmt3bb.1 MO and raised to 48 hpf. Embryos were dissociated using Ttrypsin-EDTA and passed through a 70 μM strainer to collect a single cell suspension. GFP and mApple double-positive HSPCs were isolated by forward scatter using a FACS ARIA (Becton Dickinson, Franklin Lakes, NJ). Sorted HSPC were pelleted and DNA and RNA were co-isolated using a Zymo Duet DNA/RNA miniprep kit (Zymo Research, Irvine, CA). DNA and RNA were used for bisulfite sequencing (RRBS; see below) and RNA seq analysis respectively.

## Reduced Representation by Bisulfite Sequencing (RRBS) library construction, sequencing and analysis

RRBS libraries were prepared as described previously (*Gu et al., 2011*). 62 ng of Genomic DNA was digested with MspI and TaqαI overnight. Digested DNA was purified and end-repaired and barcoded methylated adapters were ligated. DNA was size-selected using AMPure beads between 100–350 bp. The resulting DNA was bisulfite treated and PCR amplified to generate final libraries. Library quality was verified using HPLC and a bioanalyzer. We generated two separate sets of libraries and sequenced them on Illumina and SOLiD platforms. Raw single-end sequence data was trimmed for quality and adapter sequence using Trimmomatic software, trimming leading or trailing bases at quality < 5 as well as using a sliding window requiring a 4 bp average of quality > 15. Reads trimmed below 50 bp were discarded. Following trimming, reads were aligned using Bismark

software against a bisulfite converted Zv9 reference genome using non-directional alignment, allowing for up to 2 bp mismatch in the seed region. Prepared Bismark reports were parsed for regions in which coverage was $\geq$ 10x in both experimental conditions ensuring a strong comparison could be drawn. Using defined gene coordinates, regions overlapping gene bodies were combined and conversions as well as non-conversions were summed to provide an overall bisulfite conversion rate pergene. The ratio of conversions was then tested for change between conditions using a two-proportion z-test testing the hypothesis that the proportion of bisulfite conversion had altered between conditions. Reads from two independent sequencing rounds were pooled and aligned to the zebrafish genome.

## RNA sequencing and analysis

Two independent sequencing libraries were generated using SMARTer and TruSeq RNA library preparation kit. Paired-end reads were trimmed using trimmomatic and aligned to zebrafish genome using TOPHAT, differential expression analysis was performed using Cufflinks with FPKM. The PARTEK genomic suite was used to calculate differentially expressed genes by ANOVA between two control and dnmt3bb.1 deficient HSPC RNA. Absolute fold change in DNA methylation and RNA expression are shown in (*Figure 3—source data 1*) Ingenuity pathway alysis was carried out onRNA seqdata to identify major signaling pathways and networks affected in differentially expressed genes. For Dnmt3bb.1 targets, common genes with reduction in DNA methylation from RRBS and RNA seq analysis were tabulated. The RNA seq data were submitted to GEO under accession number GSE74929.

## Targeted blood-specific CpG island screen

Genes expressed by blood cells were identified in the ZFIN zebrafish gene expression database (http://zfin.org/cgi-bin/webdriver?MIval=aa-ZDB_home.apg) using search terms blood, HSCs, ventral wall of dorsal aorta, and all associated search terms. 319 genes were identified whose expression in blood had been confirmed by in situ staining or PCR. Each of these 319 gene loci were then further analyzed using the UCSC genome browser or WashU epigenetics browser to determine whether there were CpG islands in their gene bodies. The gene body was defined as 5'UTR, all exons and introns and 3' UTR. 57 genes with 72 CpG islands in their gene bodies were identified. To screen for the methylation of these 72 CpG islands, we designed bisulfite PCR primers using MethPrimer (*Li and Dahiya, 2002*): http://www.urogene.org/cgi-bin/methprimer/methprimer.cgi. To predict methylation of CpG islands by Rrestriction enzyme digestion patterns, we developed a program (http://bisulfite.nichd.nih.gov) to computationally convert all Cs to Ts and predict changes in digestion patterns using specific restriction enzymes. Using this information, we subjected PCR amplified CpG islands to digestion with the appropriate restriction enzymes and the resulting products were resolved on 2.5% Metaphor agarose. Ten CpG islands showed evidence of methylation. These ten CpG islands were subjected to bisulfite sequencing (see below) using genomic DNA prepared from control and dnmt3bb.1 morpholino-injected animals to determine whether the methylation of any of these CpG islands was dependent on runx1 function. All the PCR primers used in this analysis are given in the methods *Supplementary file 1C*.

## Genomic DNA isolation, bisulfite conversion, and sequencing

Genomic DNA was isolated from FACS sorted HSPCs or from circulating cells isolated after tail clip at day 5–7. Cells were collected in 1X PBS and genomic DNA was isolated using cell lysis buffer (10 mM Tris-HCl pH8, 5 mM EDTA, 1% SDS and 20 µg/ml Proteinase K) followed by phenol:chloroform extraction and ethanol precipitation. 1 µg of DNA was bisulfite converted using EZ DNA methylation-Lightning kit from Zymo research, following manufacturers instructions. Bisulfite converted DNA was PCR amplified using primers designed by MethPrimer (*Li and Dahiya, 2002*): http://www.urogene.org/cgi-bin/methprimer/methprimer.cgi and are listed in the methods *Supplementary file 1B*. One Taq Hotstart 2X master mix standard buffer (New England Biolabs, Ipswich, MA) or Zymo-Taq DNA polymerase (Zymo Research) enzymes were used for amplifying bisulfite converted DNA. PCR amplicons were purified and cloned using pCRII-TOPO TA cloning kit. Mini-prepped colonies were sequenced using T7 or SP6 sequencing primers. Sequencing results were analyzed using QUMA (*Kumaki et al., 2008*): http://quma.cdb.riken.jp/.

## RNA isolation, cDNA synthesis, RT-PCR and qRT-PCR

Total cellular RNA was isolated from 20–30 embryos per treatment using Trizol reagent or by Zymo Duet DNA/RNA kit. RNA was DNaseI treated to remove trace amounts of genomic DNA and phenol:chloroform extracted. Equal amounts of RNA were converted into cDNA using ThermoScript RT-PCR system (Thermo Fisher Scientific). The resulting cDNA was used for either RT-PCR or qRT-PCR using Sso Advanced SYBRGreen PCR mix (BioRad, Hercules, CA) on a CFX96 Real Time PCR system. Primers used in qRT-PCR are listed in the *Supplementary file 1B*.

## Chromatin immunoprecipitation

*Tol2*(*hsp:gfp*) or *Tol2*(*hsp:dnmt3bb.1-gfp*) constructs were injected at the one -cell stage and induced at 3 hpf by a 37°C heat shock for 30 min. Embryos were harvested at 5 hpf for CHIP analysis. CHIP was performed using anti-GFP (Developmental studies hybridoma bank: DSHB-GFP-12A6 (*Sanchez et al., 2014*) and Thermo Fisher Scientific: A11122) and anti-Methyl Cytosine (Zyme Research: A3001 and Calbiochem, San Diego, CA; Cat no: NA81) antibodies (*Nair et al., 2011*) using the EAZY CHIP chromatin immunoprecipitation kit (Millipore, Temecula, CA; Cat no 17–371) as described by Ro and Dawid (*Ro and Dawid, 2011*). Immunoprecipitated DNA was eluted using chelex beads and used in PCR analysis. PCR primers used in this study are described in *Supplementary file 1B*.

## Cell transplantation

Cell transplants were carried as described previously, with minor modifications(*Traver et al., 2003*). Donor embryos were injected at onecell stage with *Tol2*(*hsp70:gfp*) or *Tol2*(*hsp70:dnmt3bb.1-gfp*) plasmids with rhodamine- and biotin-dextran as lineage tracers. At 3 hpf, embryos were heat shocked to induce transgene expression at 37°C for 30 min. Embryos were allowed to recover at 28°C until 5 hpf. At 5 hpf, manually dechorionated embryos were lysed in yolk dissociation buffer (55 mM NaCl, 1.8 mM KCl, 1.25 mM NaHCO$_3$) and spun at 300 g for 1 min at room temperature. The cell pellet was washed twice using 1X PBS without calcium and agnesium and re-suspended in the same buffer with 5% FBS. Borosilicate needles without filaments were back-filled using cell suspension and pressure injected in the cardinal vein of a 48 hpf host embryo. Roughly, 20–25 cells were transplanted in each host embryo. Transplanted host embryos were allowed to grow until 5 dpf, then fixed and stained for biotin using Streptavidin Alexa Fluor 546 (S-11225) or Alexa Fluor 488 (S-11223) from Thermo Fisher Scientific. Embryos were imaged using an Olympus FV1000 confocal and labeled cells were counted in the thymus and in the aortic arch/heart region. We performed three independent rounds of transplants in EK WT and lck-GFP transgenic backgrounds. In total, 38 and 30 embryos were analyzed for gfp alone and dnmt3bb.1-gfp transplanted cells, respectively. Thymus localizing donor cells and heart and aortic arch localizing donor cells were counted in the 11/38 gfp alone and 17/30 dnmt3bb.1-gfp transplanted host embryos.

## Histology and blood smear

Adult zebrafish sagittal sections, H&E staining, and blood smears were prepared as described earlier (*Traver et al., 2003*; *2004*; *Sood et al., 2010*).

## Acknowledgements

We thank Drs. Igor Dawid, Janet Rossant, Jeff Gross, Sudipto Roy, Ajay Chitnis, Karuna Sampath, Mahendra Rao, Manfred Boehm and the NIH zebrafish community for sharing reagents, protocols and discussions. We thank the members of the PGD for critical comments on the manuscript and members of the Weinstein lab for technical help and critical suggestions. This research was supported by the intramural programs of the NICHD (to BMW) and NHGRI (to PPL), NIH, and an NICHD Directors Award (to AVG), by the Leducq Foundation (to BMW), and by a K99DE24190 award (to JTN).

## Additional information

### Funding

| Funder | Grant reference number | Author |
|---|---|---|
| National Institute of Child Health and Human Development | ZIA-HD001011 | Aniket V Gore<br>Brett Athans<br>Kristin Johnson<br>Daniel Castranova<br>Van N Pham<br>Matthew G Butler<br>Brant M Weinstein |
| National Human Genome Research Institute | | Erica Bresciani<br>Paul P Liu |
| Fondation Leducq | | Aniket V Gore<br>Brett Athans<br>Kristin Johnson<br>Daniel Castranova<br>Van N Pham<br>Matthew G Butler<br>Brant M Weinstein |
| National Institute of Child Health and Human Development | | James R Iben<br>Valya Russanova<br>Lisa Williams-Simons<br>Bejamin Feldman |
| National Institute of Dental and Craniofacial Research | DE13834 | Charles B Kimmel |
| National Institute of Dental and Craniofacial Research | K99DE024190 | James T Nichols |
| National Institute of Child Health and Human Development | P01HD22486 | Charles B Kimmel |

The funders had no role in study design, data collection and interpretation, or the decision to submit the work for publication.

### Author contributions

AVG, Conception and design, Acquisition of data, Analysis and interpretation of data, Drafting or revising the article; BA, KJ, VNP, LWS, Acquisition of data, Analysis and interpretation of data; JRI, PPL, BMW, Conception and design, Analysis and interpretation of data, Drafting or revising the article; VR, DC, Conception and design, Acquisition of data, Analysis and interpretation of data; MGB, BF, Acquisition of data, Analysis and interpretation of data, Drafting or revising the article; JTN, Acquisition of data, Drafting or revising the article, Contributed unpublished essential data or reagents; EB, Analysis and interpretation of data, Drafting or revising the article, Contributed unpublished essential data or reagents; CBK, Analysis and interpretation of data, Contributed unpublished essential data or reagents

### Ethics

Animal experimentation: This study was performed in strict accordance with the recommendations in the Guide for the Care and Use of Laboratory Animals of the National Institutes of Health. All of the animals were handled according to approved institutional animal care and use committee (IACUC) protocol (12-039) of the National institute of child health and human development.

## Additional files

### Supplementary files

• Supplementary file 1. (A) List of morpholinos used in the study. A list of all of the antisense morpholino oligonucleotides used in this study. (B) List of primers used in the study. A list of all of the primer pairs used for qRT-PCR, bisulfite sequencing, genotyping, cloning and CHIP analysis in this

study. (C) List of primers used in targeted blood specific CpG island screen. A list of all the primers used for the targeted blood specific CpG island screen (Analysis shown in *Figure 4*).

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
