## [Decision Letter]

Thank you for submitting your work entitled "Epigenetic regulation of hematopoiesis by DNA methylation" for consideration by *eLife*. Your article has been reviewed by three peer reviewers, and the evaluation has been overseen by Janet Rossant as the Senior Editor. One of the reviewers has agreed to reveal his identity: David Traver.

The reviewers have discussed the reviews with one another and the Reviewing editor has drafted this decision to help you prepare a revised submission.

Summary:

The reviewers were uniformly enthusiastic about the paper in terms of both the novelty of the findings and the elegance of the work. The authors elegantly describe the role of Dnmt3bb.1 in the maintenance of zebrafish hematopoietic stem and progenitor cells in the known network of Notch, Runx1 and Cmyb. They show that Runx1 drives expression of Dnmt3bb.1 and Cmyb for HSPC specification. While it was not known how Cmyb expression is maintained after Runx1 inactivation, the authors show that CpG methylation in exon 1 of Cmyb by Dnmt3bb.1 leads to continued expression of Cmyb even in the absence of Runx1. The authors also demonstrate that Dnmt3bb.1 can activate HSPC genes also in other tissues and seems to act as a general switch to activate HSPC-specific gene expression by activating Cmyb independently of endothelial genes.

While the question remains how general such switch may actually be, the function of Dnmt3bb.1 in HSPC maintenance is very exciting. The story is laid out nicely; the paper is well written and easy to read.

Essential revisions:

There were no major concerns with the paper but the following need to be addressed in a revised version.

1) In Figure 1, the authors suggest that *dnmt3bb.1* is regulated by *runx* and *notch* and is reduced in HSPCs lacking *runx* or *notch*. However, this conclusion is not well supported because HSPCs themselves are largely absent in these mutants. While the authors want to place *dnmt3bb.1* in a genetic hierarchy, these conclusions are not a major focus of the paper and should be minimized. The authors should at least clarify that loss of *dnmt3bb.1* expression in *runx1* MO, *runx* mutants, and *mib* mutants is likely because the HSPCs themselves are gone.

2) In Figure 6, are the transplanted cells in this experiment *gfp*-sorted for bulk blastula cells from injected embryos? The rationale for this experiment is unclear because the authors claim it is an assay of HSPC differentiation. However, the early embryonic cells that were transplanted could not have developed into mature HSPCs. The number of homing cells is very low and only just reaches significance compared to controls. How many replicates were quantified? If the transplanted cells are differentiated HSPCs then you would also expect to find them in the CHT and kidney at 5 dpf. Does the appearance in the thymus suggest a lymphoid bias?

---

## [Author Response]

Essential revisions:

*There were no major concerns with the paper but the following need to be addressed in a revised version.1) In Figure 1, the authors suggest that* dnmt3bb.1 *is regulated by* runx *and* notch *and is reduced in HSPCs lacking* runx *or* notch*. However, this conclusion is not well supported because HSPCs themselves are largely absent in these mutants. While the authors want to place* dnmt3bb.1 *in a genetic hierarchy, these conclusions are not a major focus of the paper and should be minimized. The authors should at least clarify that loss of* dnmt3bb.1 *expression in* runx1 *MO,* runx *mutants, and* mib *mutants is likely because the HSPCs themselves are gone.*

We thank the reviewers for these insightful comments and for the opportunity to clarify our results. We have modified our text as suggested by the reviewers to note that loss of *dnmt3b.1* expression in *runx1* MO, *runx* mutants, and *mib* mutants likely reflects at least in part the loss of HSPC cells:

We have changed “HSPC-specific expression of Dnmt3bb.1 is regulated by *notch1* and *runx1*” to “Dnmt3bb.1 is expressed in HSPC downstream from *notch1* and *runx1”.* (Abstract)

We have changed “These results show that *dnmt3bb.1* is expressed by developing HSPCs and that its expression is dependent on the established Notch-Runx1 pathway for HSPC specification” to “These results show that *dnmt3bb.1* is expressed by developing HSPCs downstream from the established Notch-Runx1 pathway for HSPC specification, and that its expression is lost when HSPC are not properly specified”. (Results)

We have added: “Since *runx1* and *mind bomb* mutant embryos have strongly reduced numbers of HSPCs, reduced *dnmt3bb.1* expression may reflect reduced HSPC numbers rather than loss of direct gene regulation”. (Discussion)

*2) In Figure 6, are the transplanted cells in this experiment* gfp*-sorted for bulk blastula cells from injected embryos?*

For the experiments in Figure 6 we transplanted approximately 20-25 dissociated but unsorted cells from 5 hpf heat-shocked donor embryos into each 48 hpf host embryo by intravenous injection. We did attempt to perform cell sorting of GFP positive blastula cells or enrichment of these cells by manual dissection before transplanting them into host embryos, but we were able to retrieve very few healthy cells for transplantation after these procedures and nearly all cells that we were able to transplant disappeared from the host embryos within 24 hrs, presumably due to cell death (the large blastula cells appear to be very fragile and easily damaged by manipulation).

The rationale for this experiment is unclear because the authors claim it is an assay of HSPC differentiation. However, the early embryonic cells that were transplanted could not have developed into mature HSPCs.

We agree with the reviewers that this is not really an assay of HSPC differentiation, but merely examining whether *dnmt3bb.1* overexpression in blastula cells promotes one specific functional property of HSPC, their ability to home to the thymus. We have reworded our text and discussion accordingly.

We changed “To further investigate the ability of *dnmt3bb.1* to drive HSPC differentiation in naive blastula cells…” to “To further investigate the ability of *dnmt3bb.1* to promote functional properties of HSPC in naive blastula cells…” (Results).

We changed “Transplanted cells from *dnmt3bb.1*-overexpressing blastulae similarly preferentially ‘home’ to the thymus and kidney (data not shown), suggesting that *dnmt3bb.1*-overexpressing cells are behaving analogously to the previously reported CD41+ HSPCs” to “Transplanted cells from *dnmt3bb.1*-overexpressing blastulae also preferentially ‘home’ to the thymus and to the kidney (data not shown), suggesting that these dnmt3bb.1-overexpressing cells are recapitulating at least one functional property of HSPCs”. (Discussion)

The number of homing cells is very low and only just reaches significance compared to controls.

Since we were not able to specifically FACS sort or dissect out healthy GFP-positive blastula cells for transplantation (see above), to increase the significance of our results we performed several additional rounds of transplantation using our whole embryo cell dissociation protocol, counting the number of cells localizing to host thymus. We present the new combined data in Figure 6 with revised numbers and a new lower p value (p < 0.001). The requested details regarding the replicates of experiments performed, numbers of embryos analyzed, etc. have been added to the Methods section.

We should also note that the counts of cells homing to the thymus carried out in Figure 6 do not take into account the hosts in the *gfp* control or *dnmt3bb.1-gfp* transplants that did *not* have cells in the thymus. In fact there were nearly twice as many host animals that received cells in the thymus in the *dnmt3bb.1-gfp* transplants (57%) compared to the *gfp* control transplants (29%). If this is taken into account, the significance of our results is even higher, but we decided not to include the “zero” class in our counts in Figure 6 in order to be conservative in our interpretation of these experiments.

How many replicates were quantified? If the transplanted cells are differentiated HSPCs then you would also expect to find them in the CHT and kidney at 5 dpf. Does the appearance in the thymus suggest a lymphoid bias?

We examined both thymus and kidney and were able to observe contribution of transplanted cells to both organs, however we chose to perform quantitative measurement of cell contribution only to the thymus because of technical difficulties that made it extremely challenging to accurately measure and/or image cell contribution to the kidney – in particular very high autofluorescence in the adjacent tissues and yolk cell that made observation of individual cells difficult in the kidney. We did not want to provide quantitative data that we did not feel 100% confident about. At this point we do not believe that there is a lymphoid bias to the contribution of these cells, although a rigorous quantitative exploration of this would be needed to draw any more definitive conclusions.